

# Polarons and bipolarons in a two-dimensional square lattice

Shanshan Ding[1], G. A. Domínguez-Castro[2,3], Aleksi Julku[1],
Arturo Camacho-Guardian[2] and Georg M. Bruun[1,4*]

**1** Center for Complex Quantum Systems, Department of Physics and Astronomy,
Aarhus University, Ny Munkegade, DK-8000 Aarhus C, Denmark
**2** Instituto de Física, Universidad Nacional Autónoma de México,
Apartado Postal 20-364, Ciudad de México C.P. 01000, Mexico
**3** Institut für Theoretische Physik, Leibniz Universität Hannover,
Appelstrasse 2, DE-30167 Hannover, Germany
**4** Shenzhen Institute for Quantum Science and Engineering and Department of Physics,
Southern University of Science and Technology, Shenzhen 518055, China

★ bruungmb@phys.au.dk

## Abstract

Quasiparticles and their interactions are a key part of our understanding of quantum many-body systems. Quantum simulation experiments with cold atoms have in recent years advanced our understanding of isolated quasiparticles, but so far they have provided limited information regarding their interactions and possible bound states. Here, we show how exploring mobile impurities immersed in a Bose-Einstein condensate (BEC) in a two-dimensional lattice can address this problem. First, the spectral properties of individual impurities are examined, and in addition to the attractive and repulsive polarons known from continuum gases, we identify a new kind of quasiparticle stable for repulsive boson-impurity interactions. The spatial properties of polarons are calculated showing that there is an increased density of bosons at the site of the impurity both for repulsive and attractive interactions. We then derive an effective Schrödinger equation describing two polarons interacting via the exchange of density oscillations in the BEC, which takes into account strong impurity-boson two-body correlations. Using this, we show that the attractive nature of the effective interaction between two polarons combined with the two-dimensionality of the lattice leads to the formation of bound states – i.e. bipolarons. The wave functions of the bipolarons are examined showing that the ground state is symmetric under particle exchange and therefore relevant for bosonic impurities, whereas the first excited state is doubly degenerate and odd under particle exchange making it relevant for fermionic impurities. Our results show that quantum gas microscopy in optical lattices is a promising platform to explore the spatial properties of polarons as well as to finally observe the elusive bipolarons.

## 1 Introduction

In seminal works, Landau, Pekar, and Fröhlich [1, 2] demonstrated that an electron moving through a dielectric distorts the surrounding crystal lattice so that it is dressed by phonons, which leads to the formation of a particle-like object coined a quasiparticle. Landau then realized that this effect is more general and that many quantum systems can be described in terms of interacting quasiparticles with properties that may be quite different from the underlying bare particles [3, 4]. This insight stands out as a highlight in theoretical physics, since it provides a simple yet accurate description of a wide range of natural systems including electrons in solids, liquid helium mixtures, nuclear matter, elementary particles, and even biological systems [3–7].

    The realisation of quasiparticles in quantum degenerate atomic gases has allowed a systematic investigation of their properties [8–15]. In these gases, the quasiparticle, called a polaron, consists of an impurity atom interacting with a surrounding atomic gas and we now have an accurate description of individual polarons formed in fermionic gases [16, 17], whereas open questions remain regarding polarons in Bose-Einstein condensates (BECs) for strong interactions. So far, these experiments have however not revealed much information regarding the

interactions between the quasiparticles, even though they have been predicted to exist both for polarons in Fermi [18–23] and Bose gases [23–27]. This is an important open question, since such interactions are an inherent property of quasiparticles as they affect each other by modulating the surrounding medium. Indeed, interactions between quasiparticles are instrumental for understanding their thermodynamic as well as dynamical properties [3, 4]. A particularly striking effect is that they can give rise to bound states. This is the origin of conventional superconductors where the interaction is mediated by phonons [28], and a quasiparticle interaction mediated by spin fluctuations is conjectured to be the mechanism behind high temperature superconductivity [29, 30]. In the case of polarons, such bound states are called bipolarons, and suggested as a mechanism for charge transport in polymers [31] as well as magnetoresistance in organic materials [32]. Bipolarons have also been predicted to exist in atomic BECs [33] but have so far not been observed.

Ultracold atoms in optical lattices represent a powerful quantum simulation platform for many-body physics since they realize the Hubbard model essentially perfectly [34]. Moreover, the ability to take pictures of individual atoms using quantum gas microscopy provides detailed spatial information of the quantum states that compliments the usual information obtained from spectroscopy [35, 36]. This has led to a range of experimental breakthroughs regarding quantum magnetism, topological matter, quantum phase transitions, and non-equilibrium physics [37, 38]. Impurities in a one-dimensional (1D) lattice containing a BEC have been considered [39–42], and the effects of the Mott-insulator to superfluid transition of bosons in a 2D square lattice on the polaron have been analyzed for weak boson-impurity interactions [43].

Here, we show that impurity atoms in a 2D optical lattice containing an atomic BEC represent a promising setup to observe the spatial properties of polarons and the formation of bipolarons for the first time. We first analyze the properties of single polarons and demonstrate that as a consequence of the lattice, a new undamped quasiparticle branch that has no analogue in continuum systems emerges for repulsive interactions. The spatial correlations between the impurity and the surrounding bosons are explored and we show that the boson density is significantly enhanced at the impurity site both for attractive and repulsive interactions. Having analysed individual polarons, we then derive an effective non-local Schrödinger equation describing the dynamics of two interacting polarons in the BEC. The interaction between the polarons is mediated by Bogoliubov modes in the BEC, and we show that it is attractive and can be strong enough to support bound states. While the wave function of the ground state bipolaron has an *s*-wave character and is symmetric under particle exchange, the first excited doubly degenerate bipolaron states have a *p*-wave character and are odd under particle exchange, making them relevant to bosonic or fermionic impurities respectively. We argue that the bipolarons can be observed via the spatial correlations between two impurities using quantum gas microscopy available in optical lattices.

## 2 System

We consider mobile impurities mixed with identical bosons in a 2D square lattice, see Fig. 1. The Hamiltonian is

$$
\begin{aligned}
\hat{H} = & -t_B \sum_{\langle \mathbf{i},\mathbf{j} \rangle} \hat{b}_{\mathbf{i}}^\dagger \hat{b}_{\mathbf{j}} + \frac{U_B}{2} \sum_{\mathbf{i}} \hat{b}_{\mathbf{i}}^\dagger \hat{b}_{\mathbf{i}}^\dagger \hat{b}_{\mathbf{i}} \hat{b}_{\mathbf{i}} - \mu_B \sum_{\mathbf{i}} \hat{b}_{\mathbf{i}}^\dagger \hat{b}_{\mathbf{i}} \\
& -t_I \sum_{\langle \mathbf{i},\mathbf{j} \rangle} \hat{c}_{\mathbf{i}}^\dagger \hat{c}_{\mathbf{j}} + \frac{U_I}{2} \sum_{\mathbf{i}} \hat{c}_{\mathbf{i}}^\dagger \hat{c}_{\mathbf{i}}^\dagger \hat{c}_{\mathbf{i}} \hat{c}_{\mathbf{i}} + U_{BI} \sum_{\mathbf{i}} \hat{b}_{\mathbf{i}}^\dagger \hat{c}_{\mathbf{i}}^\dagger \hat{c}_{\mathbf{i}} \hat{b}_{\mathbf{i}} .
\end{aligned}
\tag{1}
$$

Here, operators $\hat{b}_{\mathbf{i}}$ and $\hat{c}_{\mathbf{i}}$ remove a boson and an impurity at lattice site $\mathbf{i}$, the nearest neighbour hopping matrix element is $t_B$ for the bosons and $t_I$ for the impurities, $U_B > 0$ is the on-site

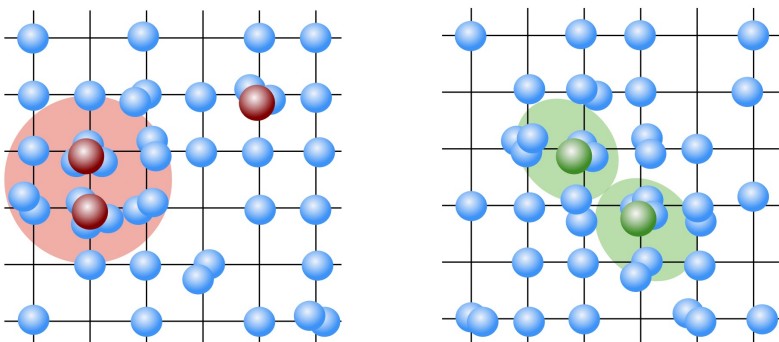

Figure 1: Illustration of the system considered. Mobile bosonic or fermionic impurities (red and green balls respectively) in a square lattice containing a BEC (blue balls). Interactions between the BEC and the impurities led to the formation of polarons, which in turn can form bound states, i.e. bipolarons, due to an induced interaction mediated by density ripples in the BEC. When the impurities are bosonic, the ground state bipolaron has $C_4$ "s-wave" symmetry whereas it has $C_2$ "p-wave" symmetry for fermionic impurities.

boson-boson repulsion, $U_I$ is the on-site impurity-impurity interaction, and $U_{BI}$ is the on-site interaction between the bosons and the impurities. The chemical potential of the bosons is $\mu_B$ and we set $\hbar$ and the lattice constant to unity.

The bosons form a BEC with $n_0$ particles per lattice site, and we assume that the interaction $U_B$ is weak so that the BEC is accurately described by Bogoliubov theory. This gives the chemical potential $\mu_B = -4t_B + n_0 U_B$ and the excitation spectrum $E_\mathbf{k} = \sqrt{\epsilon_{B\mathbf{k}}^2 + 2\epsilon_{B\mathbf{k}} n_0 U_B}$, where $\epsilon_{B\mathbf{k}} = -2t_B(\cos k_x + \cos k_y) + 4t_B$. Here, $\mathbf{k}$ is the crystal momentum of the excitation within the first Brillouin zone (BZ) of the lattice. In the rest of the paper, we take for concreteness $t_I = t_B$.

## 3 Two-body scattering and bound states

We first study the scattering of an impurity and a boson in an empty lattice. For the contact interaction $U_{BI}$, the scattering matrix only depends on the total center-of-mass momentum $\mathbf{P}$ and the energy $\omega$ of the pair and takes the simple form

$$\mathcal{T}(\mathbf{P}, \omega) = \frac{U_{BI}}{1 - U_{BI}\Pi_\mathrm{v}(\mathbf{P}, \omega)}. \tag{2}$$

Here,

$$\Pi_\mathrm{v}(\mathbf{P}, \omega) = \frac{1}{M} \sum_\mathbf{k} \frac{1}{\omega - \epsilon_{B\mathbf{P}/2+\mathbf{k}} - \epsilon_{I\mathbf{P}/2-\mathbf{k}}} \tag{3}$$

is the pair propagator in an empty lattice with $\epsilon_{I\mathbf{k}} = -2t_I(\cos k_x + \cos k_y)$ and $M$ the number of lattice sites. We measure energies with respect to the minimum $-4t_B$ of the tight-binding band of the bosons in order to facilitate comparisons with the case when a BEC is present and the energies are measured with respect to $\mu_B$. For an infinite lattice, the pair propagator in Eq. (3) can be evaluated analytically giving

$$\Pi_\mathrm{v}(\mathbf{0}, \omega)t_B = \begin{cases} \left[\mathrm{sgn}(z)K(|z|) - iK(\sqrt{1-z^2})\right]/4\pi, \\ K(|z|^{-1})/4\pi z, \end{cases} \tag{4}$$

where $K(z)$ is the complete elliptic integral of the first kind, the top line is for $-1 < z < 1$, and the bottom line is for any other case. We have defined $z = (\omega - 4t_B)/8t_B$ representing the two-body energy measured in units of the bandwidth of single particle $8t_B$. Details of deriving Eq. (4) are given in App. A. The non-zero imaginary part of $\Pi(\mathbf{0}, \omega)$ for $-4t_B < \omega < 12t_B$ comes from the continuum of impurity and boson single-particle states with zero center of mass (COM) momentum and energies being $\epsilon_{B\mathbf{k}} + \epsilon_{I-\mathbf{k}}$. Discrete poles of the scattering matrix on the other hand give the energies of any bound states consisting of one boson and one impurity in the lattice.

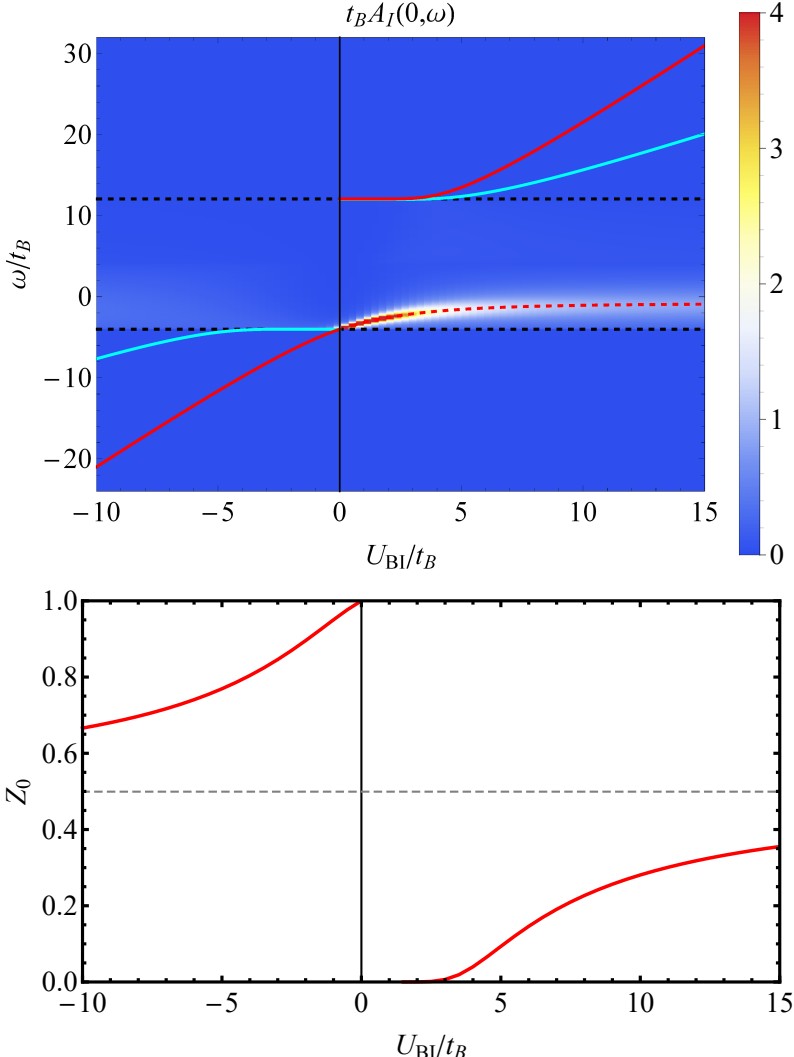

Figure 2: The top panel shows the bound state energies (cyan lines) of one impurity and one boson in an otherwise empty lattice as a function of $U_{BI}$. It also shows the spectral function of the impurity with the red solid lines indicating the energy of an undamped attractive ($U_{BI} < 0$) and upper polaron ($U_{BI} > 0$). The dashed red line gives the energy of the damped repulsive polaron, which is inside the single particle-Bogoliubov mode continuum indicated by the horizontal black dashed lines. The bottom panel shows the corresponding quasiparticle residues, where the horizontal gray dashed line shows the asymptotic value of the residue for $|U_{BI}| \to \infty$ given by Eq. (8a).

The top panel of Fig. 2 shows the energy $\epsilon_B$ of such bound states as a function of $U_{BI}$ (cyan lines). For an attractive impurity-boson interaction $U_{BI} < 0$ there is a bound state with energy below the single particle continuum that has energies between the black dashed horizontal lines. This is the lattice analogue of the bound dimer state for homogeneous atomic gases in the BEC-BCS cross-over [44]. Figure 2 furthermore shows that there is a second bound state for repulsive interaction $U_{BI} > 0$ with an energy above the single particle continuum. This bound state is stable since it has no available decay channels as the single particle continuum is bounded from above [45,46]. Such repulsively bound states have been observed for bosons in an optical lattice [47]. As shown in App. B, it follows from the two-dimensionality of the lattice that there is a two-body bound state for any non-zero value of $U_{BI}$ either above ($U_{BI} > 0$) or below ($U_{BI} < 0$) the single-particle continuum. For large $\omega$, we have $\Pi_v(\mathbf{0}, \omega) \simeq 1/8zt_B$ so that $\epsilon_B \to U_{BI} + 4t_B$ for $|U_{BI}| \to \infty$.

## 4 Polarons

We now analyse the properties of a single impurity in the BEC and the formation of quasiparticles. First, we focus on the spectral properties of the impurity, and then we discuss the spatial correlations between the impurity and the bosons.

### 4.1 Spectral properties

The spectral properties of an impurity with crystal momentum $\mathbf{k}$ immersed in the BEC can be described by the impurity Green's function

$$G_I(i\omega, \mathbf{k}) = \frac{1}{i\omega - \epsilon_{I\mathbf{k}} - \Sigma(i\omega, \mathbf{k})}, \tag{5}$$

where $\Sigma(i\omega, \mathbf{k})$ is its self-energy. The energy $\omega_{\mathbf{k}}$ of a quasiparticle is given by the pole of $G_I(i\omega, \mathbf{k})$, that is $\omega_{\mathbf{k}} - \epsilon_{I\mathbf{k}} - \mathrm{Re}\Sigma(\omega_{\mathbf{k}} + i\eta, \mathbf{k}) = 0$ ($\eta$ is a positive infinitesimal number) and the corresponding quasiparticle residue is $Z_{\mathbf{k}} = 1/[1 - \partial_\omega \mathrm{Re}\Sigma(\omega + i\eta, \mathbf{k})]\big|_{\omega = \omega_{\mathbf{k}}}$.

We use the ladder approximation to calculate the impurity self-energy [48], which has turned out to be in good agreement with experiments both for equilibrium [13,49] and non-equilibrium [50] properties of the Bose polaron in continuum systems. In this approximation, the self-energy is

$$\Sigma(i\omega, \mathbf{k}) = n_0 \mathcal{T}(i\omega, \mathbf{k}), \tag{6}$$

where the scattering matrix is given by Eq. (2) with the pair propagator generalised from Eq. (3) to take into account the presence of the BEC. This yields

$$\Pi(\mathbf{P}, i\Omega) = \frac{1}{M} \sum_{\mathbf{k}} \frac{u_{\mathbf{k}}^2}{i\Omega - \epsilon_{I\mathbf{P}-\mathbf{k}} - E_{\mathbf{k}}} \tag{7}$$

at zero temperature, where $u_{\mathbf{k}}^2 = [(\epsilon_{B\mathbf{k}} + n_0 U_B)/E_{\mathbf{k}} + 1]/2$. Details of deriving Eq. (7) are given in App. C.

In the upper panel of Fig. 2, we plot the spectral function $A_I(\mathbf{k}, \omega) = -2\mathrm{Im}G_I(\mathbf{k}, \omega + i\eta)$ of the impurity for vanishing crystal momentum $\mathbf{k} = 0$ as a function of $U_{BI}$. Here and in the rest of the paper we take the value $n_0 = 1$ and $U_B/t_B = 0.07$ for the numerical calculations. This value is experimentally realistic and ensures that the system is far away from the transition to the Mott phase [51,52]. Figure 2 shows that there is a well-defined quasiparticle branch below the dimer state energy for $U_{BI} < 0$. Its quasiparticle residue decreases from unity with decreasing $U_{BI}$ as shown in the lower panel of Fig. 2. The energy of this quasiparticle branch smoothly

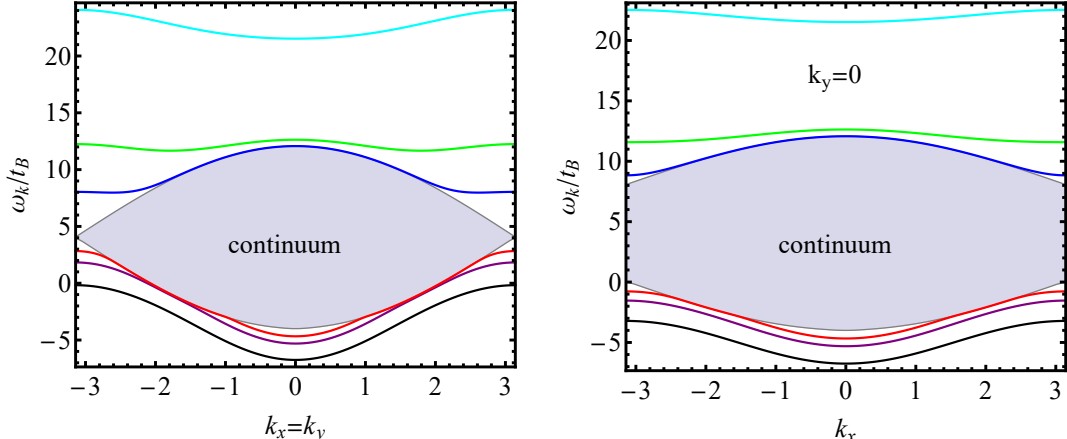

Figure 3: The energies of the attractive polaron for $U_{BI} < 0$ and upper polaron for $U_{BI} > 0$ as a function of the crystal momentum along the diagonal $k_x = k_y$ (left panel) and along the $k_x$-axis with $k_y = 0$ (right panel). The lines from bottom to top are for $U_{BI}/t_B = -2.1$ (black), $U_{BI}/t_B = -1.1$ (purple), $U_{BI}/t_B = -0.6$ (red), $U_{BI}/t_B = 2$ (blue), $U_{BI}/t_B = 4.1$ (green), and $U_{BI}/t_B = 10$ (cyan). The single particle-Bogoliubov mode continuum is indicated by the gray region.

increases to positive values when $U_{BI} > 0$, where it is broadened due to decay into the single particle-Bogoliubov mode continuum with energies $\epsilon_{I\mathbf{k}} + E_{-\mathbf{k}}$, which is almost indistinguishable from the single particle continuum in Fig. 2. This quasiparticle branch is the lattice analogue of the attractive polaron in continuum gases that smoothly evolves into a damped repulsive polaron for positive impurity-boson interaction. We note that the broadening shown in Fig. 2 may be an artefact of the approximation used. Indeed, a self-consistent approach would move the single particle-Bogoliubov mode continuum above the energy of the repulsive polaron for $\mathbf{k} = 0$ [53].

Figure 2 furthermore shows a second quasiparticle branch above the single particle-Bogoliubov mode continuum for repulsive impurity-boson interaction. It has an energy *above* that of the repulsive dimer and a residue increasing from zero with $U_{BI}$, as can be seen in the lower panel of Fig. 2. We denote this quasiparticle branch with infinite lifetime, which has no analogue for continuum gases, as the *upper polaron*. For large frequencies, we can approximate the pair propagator by $\Pi(\omega) = A/\omega$ with $A = \sum_{\mathbf{k}} u_{\mathbf{k}}^2/M$ a coefficient determined by the medium. This in turn gives the residue and energy

$$Z_{\mathbf{k}} \to \frac{n_0}{n_0 + A}, \tag{8a}$$

$$\omega_{\mathbf{k}} \to (n_0 + A)U_{BI} \tag{8b}$$

of the attractive and upper polaron for $U_{BI} \to -\infty$ and $U_{BI} \to \infty$ respectively. Equation (8a) shows that for $|U_{BI}| \to \infty$ the residue of the polaron saturates to a non-zero value that can be tuned by changing the density of the BEC. This value is shown as a horizontal dashed line in Fig. 2 (bottom). For an ideal BEC with density $n_0 = 1$, Eq. (8a) predicts $Z_{\mathbf{k}} \to 1/2$ whereas we obtain $Z_{\mathbf{k}} \to 0.4993$ for $U_B/t_B = 0.07$.

In Fig. 3, we plot the polaron energy $\omega_{\mathbf{k}}$ as a function of the crystal momentum along the diagonal $k_x = k_y$ (left panel) and along the $k_x$-axis with $k_y = 0$ (right panel) in the BZ for different values of the impurity-boson interaction $U_{BI}$. The single particle-Bogoliubov mode continuum with energies in the interval $\min_{\mathbf{q}}(\epsilon_{I\mathbf{q}} + E_{\mathbf{k}-\mathbf{q}}) \leq \omega \leq \max_{\mathbf{q}}(\epsilon_{I\mathbf{q}} + E_{\mathbf{k}-\mathbf{q}})$ is also shown. When $U_{BI} < 0$, we plot the energy of the attractive polaron, and when $U_{BI} > 0$ we

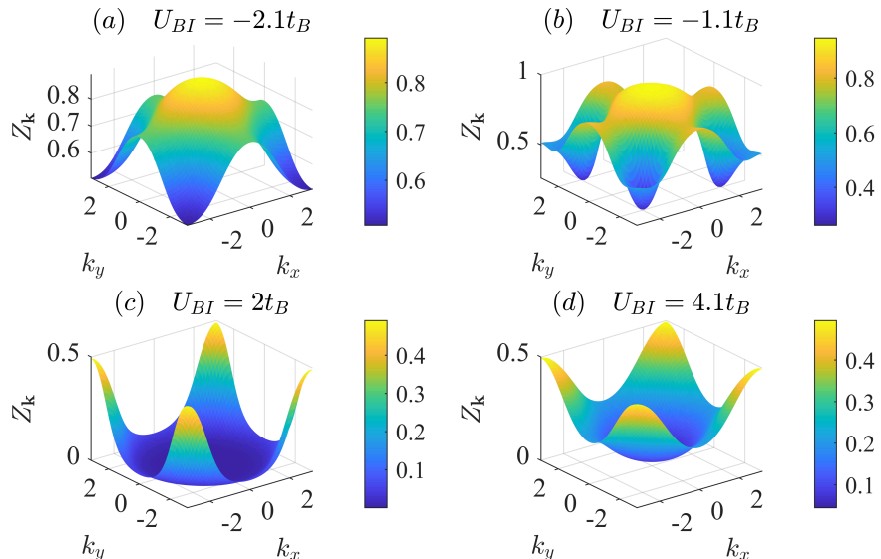

Figure 4: The quasiparticle residue of the attractive polaron for $U_{BI}/t_B = -2.1$ and $U_{BI}/t_B = -1.1$, and for the upper polaron for $U_{BI}/t_B = 2$ and $U_{BI}/t_B = 4.1$ in the BZ.

plot the energy of the upper polaron. Figure 3 shows that the energy band of the attractive polaron below the continuum has a minimum at $\mathbf{k} = 0$, whereas the energy band of the upper polaron has a more complicated structure with a local maximum $\mathbf{k} = 0$ that evolves into a minimum for increasing impurity-boson repulsion.

Figure 4 shows the quasiparticle residue of the polaron in the BZ for different values of $U_{BI}$. When $U_{BI}/t_B = -2.1$ shown in panel (a), the residue has a maximum at $\mathbf{k} = 0$ and minima at the corners of the BZ, and it is larger than $1/2$ for all $\mathbf{k}$. For the smaller attractive interaction $U_{BI}/t_B = -1.1$ shown in panel (b), the minima have moved to $k_x = k_y \simeq 2$ and correspond to a small residue, reflecting that the energy of the attractive polaron is close to the single particle-Bogoliubov mode continuum. Contrary to this, the residue of the upper polaron has maxima at the corners of the BZ whereas it is very small close to zero momentum as can be seen in panel (c) for $U_{BI}/t_B = 2$. By comparing with panel (d), we see that the residue of the upper polaron increases and is non-zero in the whole BZ for larger impurity-boson repulsion.

## 4.2 Spatial impurity-boson correlations

Inspired by the impressive single site resolution microscopy available in optical lattices [35,36], we now turn to the spatial correlations between the impurity and the bosons in the polaron states.

The spatial properties are most straightforwardly calculated from the polaron wave function, which in the ladder approximation reads

$$|\Psi_P\rangle = (\phi_0 \hat{c}^\dagger_{\mathbf{k}=0} + \sum_{\mathbf{k}} \psi_{\mathbf{k}} \hat{c}^\dagger_{\mathbf{k}} \hat{\beta}^\dagger_{-\mathbf{k}})|\text{BEC}\rangle. \qquad (9)$$

Here $\hat{c}^\dagger_{\mathbf{k}}/\hat{\beta}^\dagger_{\mathbf{k}}$ creates an impurity/Bogoliubov mode with crystal momentum $\mathbf{k}$, $|\text{BEC}\rangle$ denotes the ground state of the BEC defined by $\hat{\beta}_{\mathbf{k}}|\text{BEC}\rangle = 0$ and we have taken the momentum of the polaron to be zero. The variational parameters $\phi_0$ and $\psi_{\mathbf{k}}$ are determined by minimising the energy $\langle \Psi_P|\hat{H}|\Psi_P\rangle$, and one can show that they are directly obtained from the ladder calculation as $\phi_0 = \sqrt{Z_0}$ and $\psi_{\mathbf{k}} = \sqrt{Z_0} u_{\mathbf{k}}(\omega_0 - \epsilon_{I0})/\sqrt{nM}(\omega_0 - \epsilon_{I\mathbf{k}} - E_{\mathbf{k}})$, see App. D.

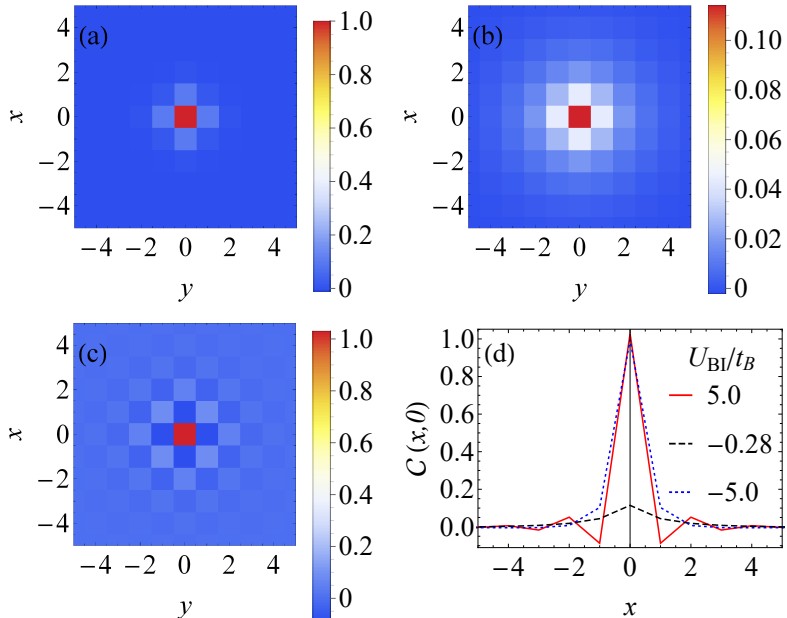

Figure 5: The spatial correlations between the impurity and the bosons as described by $\mathcal{C}(\mathbf{i})$ in the attractive polaron state for $U_{BI}/t_B = -5$ (a) and $U_{BI}/t_B = -0.28$ (b), and in the upper polaron state for $U_{BI}/t_B = 5.0$ (c). Panel (d) shows $\mathcal{C}(x,0)$ for the three impurity-boson interaction strengths.

To analyse the spatial correlations between the impurity and the surrounding BEC, we calculate

$$\mathcal{C}(\mathbf{i}-\mathbf{j}) = M \langle \Psi_P | [\hat{n}_B(\mathbf{i}) - n] \hat{n}_I(\mathbf{j}) | \Psi_P \rangle, \tag{10}$$

where $\hat{n}_B(\mathbf{i}) = \hat{b}_{\mathbf{i}}^\dagger \hat{b}_{\mathbf{i}}$ and $\hat{n}_I(\mathbf{i}) = \hat{c}_{\mathbf{i}}^\dagger \hat{c}_{\mathbf{i}}$, $n = \langle \Psi_P | \hat{n}_B(\mathbf{i}) | \Psi_P \rangle$ gives the total number of bosons at lattice site $\mathbf{i}$, and we have assumed an infinite lattice. The correlation function $\mathcal{C}(\mathbf{i}-\mathbf{j})$ gives the number of bosons at site $\mathbf{i}$ given that the impurity is at site $\mathbf{j}$. As detailed in App. D, a straightforward calculation using Bogoliubov theory yields

$$\mathcal{C}(\mathbf{i}) = \sqrt{\frac{n}{M}} \sum_{\mathbf{k}} \left[ \phi_0^* \psi_{\mathbf{k}} (u_{\mathbf{k}} + v_{\mathbf{k}}) e^{i\mathbf{k}\cdot\mathbf{i}} + \text{h.c} \right] + \frac{1}{M} \sum_{\mathbf{k},\mathbf{k}'} \psi_{\mathbf{k}}^* \psi_{\mathbf{k}'} (u_{\mathbf{k}} u_{\mathbf{k}'} + v_{\mathbf{k}} v_{\mathbf{k}'}) e^{i(\mathbf{k}-\mathbf{k}')\cdot\mathbf{i}}. \tag{11}$$

Figure 5 shows the spatial correlation function $\mathcal{C}(\mathbf{i})$ for different values of $U_{BI}$. First, for a strong and attractive impurity-boson interaction $U_{BI}/t_B = -5$ shown in panel (a), the number of bosons at the lattice site of the impurity is increased by one in the attractive polaron state, whereas it quickly relaxes to the background value away from the impurity. This corresponds to a strongly localised dressing cloud containing approximately one boson on top of the impurity. As expected, the number of bosons in the dressing cloud decreases and it becomes more spatially extended with decreasing attraction as can be seen in panel (b) plotting $\mathcal{C}(\mathbf{i})$ for $U_{BI}/t_B = -0.28$. For a strong repulsive interaction $U_{BI}/t_B = 5$ shown in panel (c), the dressing cloud is again localised at the impurity and contains approximately one boson. This is because it is energetically forbidden for the boson to tunnel to a neighbouring site in the upper polaron state. Finally, we plot in panel (d) $\mathcal{C}(x,0)$ for the three different bose-impurity interaction strengths, illustrating further how the dressing cloud becomes more localised both for large attractive and repulsive impurity-boson interaction. In addition, we see that $\mathcal{C}(x,0)$ is negative at the sites neighbouring the impurity when $U_{BI}/t_B = 5$. Physically, this means that while the impurity attracts a boson to its own lattice site in the upper polaron state, it pushes

them away from its nearest neighbour sites contrary to the case of the attractive polaron where $\mathcal{C}(\mathbf{i})$ is positive everywhere.

# 5 Mediated interaction and effective Schrödinger equation

Having analysed individual polarons, we are now ready to explore how they interact by the exchange of Bogoliubov modes, and to derive an effective Schrödinger equation for two polarons in the presence of the BEC.

## 5.1 Schrödinger equation

It is convenient to derive an effective Schrödinger equation to describe the effects of a general interaction between two polarons. To do this, we consider the scattering of two impurities with momenta and energies $p_1$ and $p_2$ to momenta and energies $p_1+q$ and $p_2-q$. The corresponding scattering matrix obeys the Bethe-Salpeter equation, which in the ladder approximation reads

$$\Gamma(p_1, p_2; q) = V(p_1, p_2; q) - \frac{T}{M} \sum_{q_1} V(p_1, p_2; q_1)$$
$$\times G_I(p_1 + q_1) G_I(p_2 - q_1) \Gamma(p_1 + q_1, p_2 - q_1; q - q_1). \tag{12}$$

Here, $V(p_1, p_2; q_1)$ is the interaction between the impurities, $T$ the temperature, and the sum is over momenta in the BZ and Matsubara frequencies. Since an interaction mediated by Bogoliubov modes is not Galilean invariant and the speed of sound in the BEC is finite, $V(p_1, p_2; q_1)$ in general depends on all momenta and energies. In order to simplify the problem, we therefore make some approximations. First, we use a pole expansion for the impurity Green's function around the polaron energies $\omega_{\mathbf{k}}$ writing $G_I(i\omega, \mathbf{k}) \simeq Z_{\mathbf{k}}/(i\omega - \omega_{\mathbf{k}})$ and multiply both sides of Eq. (12) by $Z_{\mathbf{p}_1} Z_{\mathbf{p}_2}$. This approximation corresponds to going to a quasiparticle picture with $\Gamma_{\text{eff}}(p_1, p_2; q) = Z_{\mathbf{p}_1} Z_{\mathbf{p}_2} \Gamma(p_1, p_2; q)$ the scattering matrix between two polarons with a quasiparticle interaction $V_{\text{qp}}(p_1, p_2; q) = Z_{\mathbf{p}_1} Z_{\mathbf{p}_2} V(p_1, p_2; q)$ [54]. Second, we neglect the dependence of the induced interaction on the frequency transfer, which is accurate when the binding energy of the bipolaron is small compared to the typical excitation energy of the Bogoliubov modes. Finally, we take the energy of the scattering particles to be a constant determined by the energy of the incoming polarons. Using these approximations, which are reliable as long as there are well-defined polarons with a large spectral weight and the bi-polarons are weakly bound, the frequency sum in Eq. (12) can now be evaluated analytically. The result is a Lippmann-Schwinger equation that in turn is equivalent to an effective Schrödinger equation for two polarons in the BEC as explained in App. E. Taking the COM momentum of the two polarons to be zero, the Schrödinger equation reads

$$(E_B - 2\omega_{\mathbf{k}})\psi(\mathbf{k}) = \frac{1}{M} \sum_{\mathbf{k}'} V_{\text{qp}}(\mathbf{k}, \mathbf{k}')\psi(\mathbf{k}'). \tag{13}$$

Here, $E_B$ is the energy of the two-polaron state with wave function $\psi(\mathbf{k})$, where $\mathbf{k}$ is their relative momentum. We have changed notation so that $V_{\text{qp}}(\mathbf{k}, \mathbf{k}')$ denotes the interaction between two polarons with zero COM momentum scattering from relative momentum $\mathbf{k}$ to $\mathbf{k}'$. Since the interaction depends on both the in-coming and out-going relative momentum, it is non-local reading $\sum_{\mathbf{i}'} V(\mathbf{i}, \mathbf{i}')\psi(\mathbf{i}')/M$ in real space. This is typical of effective two-body Schrödinger equations in many-body systems such as for instance the Skyrme force in nuclear matter [5].

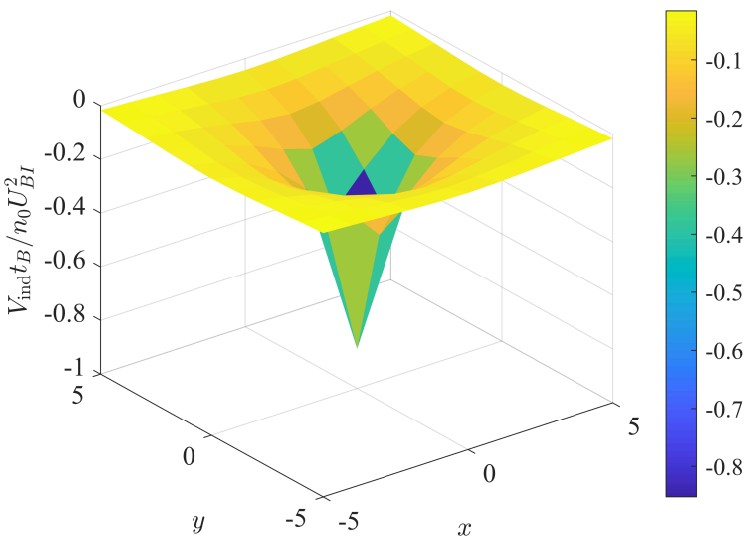

Figure 6: The induced interaction $V_{\text{ind}}(\mathbf{i})$ between two polarons in real space obtained by Fourier transforming Eq. (16).

## 5.2 Polaron-polaron interaction

There are two contributions to the quasiparticle interaction between two polarons, which we write as

$$V_{\text{qp}}(\mathbf{k}, \mathbf{k}') = Z_{\mathbf{k}}^2 U_I + V_{\text{ind}}(\mathbf{k}, \mathbf{k}'). \tag{14}$$

The first term comes from the direct interaction between the impurities, which is reduced by a factor $Z_{\mathbf{k}}^2$ since this is the impurity component of the two polarons. The second term is the induced interaction between the polarons mediated by Bogoliubov modes in the BEC. As shown in App. E, this interaction is given by

$$\begin{aligned} V_{\text{ind}}(\mathbf{k}, \mathbf{k}') = Z_{\mathbf{k}}^2 n_0 [ & 2\mathcal{T}(\mathbf{k}, \omega) G_{11}(\mathbf{k}' - \mathbf{k}, 0) \mathcal{T}(\mathbf{k}', \omega) \\ & + \mathcal{T}^2(\mathbf{k}', \omega) G_{12}(\mathbf{k}' - \mathbf{k}, 0) + \mathcal{T}^2(\mathbf{k}, \omega) G_{12}(\mathbf{k}' - \mathbf{k}, 0) ], \end{aligned} \tag{15}$$

to leading order in the number of Bogoliubov modes. Here, $G_{ij}(\mathbf{k}, 0)$ are the Green's functions of the BEC, and $\omega$ is taken to be the energy of the interacting quasiparticles. The scattering matrices in Eq. (15) take the two-body impurity-boson correlations into account exactly.

For weak impurity-boson interaction, we have $\mathcal{T} \simeq U_{BI}$ and the induced interaction simplifies to

$$V_{\text{ind}}(\mathbf{k}, \mathbf{k}') = -\frac{2n_0 U_{BI}^2 \epsilon_{B\mathbf{k}-\mathbf{k}'}}{E_{\mathbf{k}-\mathbf{k}'}^2}. \tag{16}$$

This shows that it to leading order is proportional to $U_{BI}^2$. In Fig. 6, we plot the induced interaction in real space obtained by Fourier transforming Eq. (16). This illustrates the attractive nature of the induced interaction and that its range is a few lattice sites. Approximating the single particle dispersion at small momenta as quadratic and Fourier transforming Eq. (16) to real space yields the long range behaviour $V_{\text{ind}}(r) \propto \exp(-\sqrt{2n_0 U_B / t_B} r)/\sqrt{r}$.

While we have employed a number of approximations in deriving the Schrödinger equation Eq. (13) with the quasiparticle interaction given by Eqs.(14)-(15), we note that a similar approach turns out to be remarkably accurate when compared with Monte-Carlo calculations for continuum quantum gases, even for strong impurity-boson interactions leading to large bipolaron binding energies [33].

## 6 Bipolarons

We can now address question whether the induced interaction is strong enough to lead to bound states between two polarons. Note that while the condition for two-body bound states in a 2D continuum system with a local interaction $V(r)$ is $\int dr r V(r) < 0$ [55], the present case is more complicated since we are dealing with a non-local effective interaction coming from integrating out the bosonic degrees of freedom in a lattice BEC. We focus on bound states between two attractive polarons taking $U_{BI} < 0$, and we therefore use the attractive polaron energies $\omega_{\mathbf{k}}$ in Eq. (13). Likewise, the quasiparticle interaction is calculated from Eq. (15) using the residues $Z_{\mathbf{k}}$ of the attractive polarons and the ground state energy $\omega = \omega_{\mathbf{k}=0}$.

In the top panel of Fig. 7, we plot the eigenvalue spectrum obtained from solving Eq. (13) as a function of $U_{BI}$. The bare impurities are taken to be non-interacting with $U_I = 0$ so that any bound states arise exclusively from the induced interaction. The solutions of Eq. (13) fall in two classes. First, there is a set of scattering states with energies close to those of two free attractive polarons $\omega_{\mathbf{k}} + \omega_{-\mathbf{k}}$, which form a continuum in the thermodynamic limit. Since we have subtracted the ground state energy $2\omega_{\mathbf{k}=0}$ of two unbound attractive polarons, this continuum starts at zero energy in Fig. 7. Second, we see that for sufficiently large $|U_{BI}|$ a solution with a discrete energy separated from this continuum emerges. This solution corresponds to a bound dimer state, i.e. a bipolaron, with a binding energy increasing with $|U_{BI}|$. The lower panel of Fig. 7 plots the corresponding wave function $\psi_A(\mathbf{k})$ in momentum space for $U_{BI}/t_B = -2.1$. It is centered in the BZ and is $C_4$ ("$s$-wave") symmetric due to the square lattice. We denote this bipolaron with the label "A" corresponding to the irreducible representation of the lattice symmetry group $C_{4v}$ it spans [56]. Since its wave function is symmetric under particle-exchange, i.e. a $C_2$ operation, it is relevant for bosonic impurities, whereas two fermionic impurities cannot occupy this bipolaron state. Note that the fact that the wave function is centered at the origin in momentum space makes the approximation using the ground state polaron energy $\omega_{\mathbf{k}=0}$ in the mediated interaction consistent. In Fig. 7, we also plot the probability $|\psi_A(\mathbf{r})|^2$ of finding the two impurities separated by the vector $\mathbf{r}$ in the lattice, which is obtained by a Fourier transform of $\psi_A(\mathbf{k})$. The probability is strongly localised, which means that the two impurities are mostly at the same or neighboring lattice sites when bound together in the bipolaron state.

Figure 7 shows that another negative energy branch emerges for $U_{BI}/t_B \lesssim -1.6$. This branch in fact corresponds to two degenerate bipolaron states with wave functions $\psi_{E_1}$ and $\psi_{E_2}$ plotted in the lower panel of Fig. 7 for $U_{BI}/t_B = -2.1$. These two degenerate states span the two-dimensional irreducible representation E of $C_{4v}$, which is the only one that is odd under particle exchange. These two bipolaron states are therefore relevant for fermionic impurities in the BEC [56], and their wave functions have a "$p$-wave" symmetry with a node at the origin. As a result, they are considerably more extended in real space as compared to the lowest bipolaron state, meaning that the two impurities are most likely to be at neighbouring lattice sites whereas they are never at the same site. Note that one can form linear combinations of these two wave functions that are extended along the $x$- and $y$-axis instead of along the diagonals.

These results demonstrate that the mediated interaction between two polarons is indeed strong enough to support bound states in the 2D lattice. The bipolarons should furthermore be observable by looking at the spatial correlations between two impurities using the single site resolution of atomic microscopy in optical lattices.

To explore the effects of a non-zero direct interaction between the bare impurities, we plot in Fig. 8 the energy of the ground state bipolaron with A symmetry as a function of $U_I$ for $U_{BI}/t_B = -2.1$. The dashed line shows the binding energy of the two impurities in an empty lattice calculated from the pole of Eq. (2) with $U_{BI}$ replaced by $U_I$. As discussed in Sec. 3,

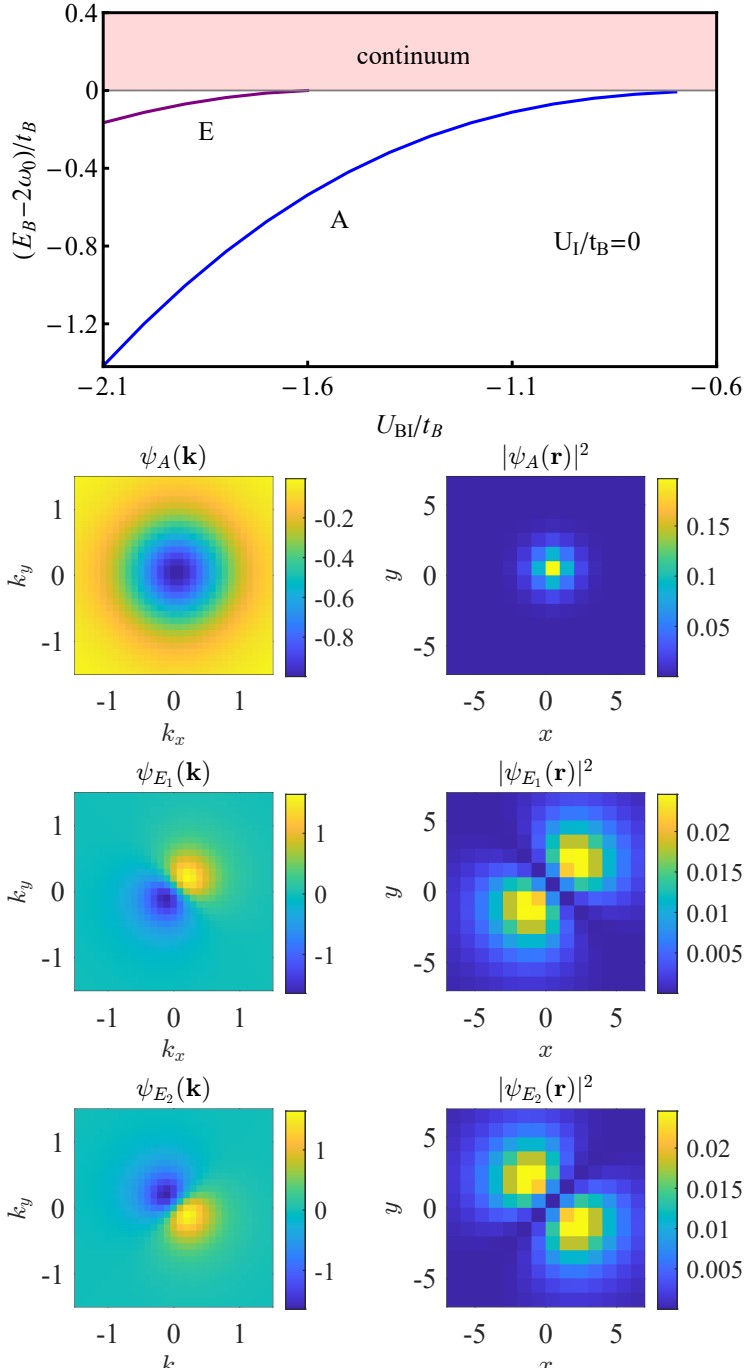

Figure 7: The top panel shows the energy of the bipolaron states relative to the continuum of two unbound attractive polarons with zero COM momentum as a function of $U_{BI}$ for $U_I = 0$. The blue line (A) is the energy of the $C_4$ symmetric bipolaron relevant for bosonic impurities whereas the purple line (E) gives the energy of the two bipolaron states with $C_2$ symmetry relevant for fermionic impurities. The bottom panel shows the corresponding wave functions in momentum space and densities in real space for $U_{BI}/t_B = -2.1$.

there is always one bound state below the continuum for $U_I < 0$ due to the 2D nature of the lattice [45]. Figure 8 shows that the presence of the BEC increases the binding energy of the bipolaron for $U_I/t_B \gtrsim -11.7$ making it stable also for $U_I > 0$. This is due to the attractive

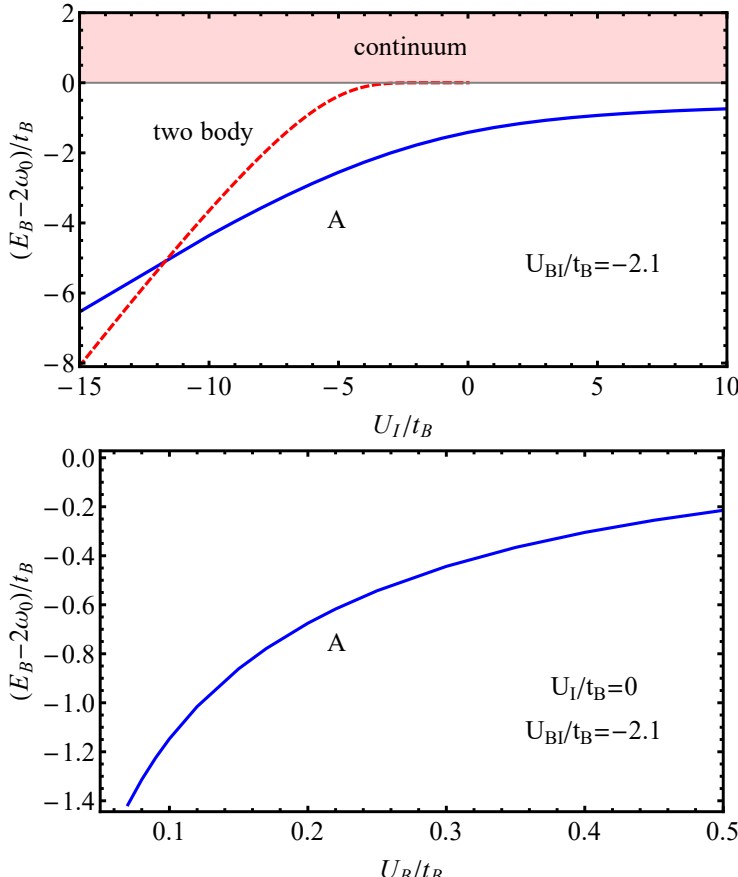

Figure 8: The top panel shows the energy of the ground state bipolaron (blue solid) relative to the continuum of two unbound attractive polarons with zero COM momentum as a function of the impurity-impurity interaction strength $U_I$ for $U_{BI}/t_B = -2.1$. The dashed red line is the bound state energy of two impurities in an empty lattice. The bottom panel shows the energy of the ground state bipolaron as a function of the boson-boson repulsion $U_B$ for $U_{BI}/t_B = -2.1$ and $U_I = 0$.

interaction mediated by the BEC. Somewhat surprisingly however, the bipolaron is less bound for $U_I/t_B \lesssim -11.7$. This is because the BEC decreases the impurity component of the polarons as quantified through their residues $Z_{\mathbf{k}}$, which in turn decreases the attraction coming from the direct interaction between the bare impurities, as can be seen explicitly from the first term in Eq. (14). Note that there are no excited dimer states with E symmetry in an empty lattice as is easily understood from the fact they have a node in the origin making them insensitive to an on-site interaction. Hence, a bipolaron containing two identical fermionic impurities is *only* stable in the presence of a BEC.

We finally explore how the binding energy of the bipolaron depends on the repulsion between the bosons in the BEC. In the lower panel of Fig. 8, the ground state bipolaron energy is plotted as a function of $U_B$ with $U_I = 0$, and $U_{BI}/t_B = -2.1$. The binding energy increases with decreasing boson-boson repulsion. This can be understood from the perturbative expression Eq. (16) showing that the induced interaction increases with decreasing $U_B$. Physically it reflects that the BEC becomes more compressible with decreasing boson-boson repulsion making the induced interaction stronger. One should however note that our theory neglects retardation effects and therefore likely will break down as the speed of sound in the BEC decreases for $U_B \to 0$. We expect retardation effects to decrease the binding energy of the bipolaron.

# 7 Conclusions and Outlook

We explored mobile impurities immersed in a BEC in a square lattice. By calculating the spectral properties of the impurities, we showed that the lattice gives rise to a new and stable polaron branch above the single particle continuum, which is not present for continuum systems. We derived an expression for the induced interaction between two polarons mediated by the exchange of density oscillations in the BEC, which takes into account strong impurity-boson correlations. Using this induced interaction in an effective Schrödinger equation for two polarons, we showed that it can be strong enough to bind two polarons into a bipolaron. The wave function of the ground state bipolaron was shown to be symmetric under particle exchange and therefore relevant for bosonic impurities, whereas the doubly degenerate first excited bipolaron states are odd making them relevant for fermionic impurities. We investigated the spatial correlations between the bosons and the impurity in the polaron states, as well as between two impurities in the bipolaron states and showed that both should be observable using high resolution quantum gas microscopy in optical lattices. Our results show that optical lattices are a promising platform to explore new aspects of polaron physics such as their spatial properties, and for observing the elusive bipolarons for the first time in quantum degenerate gases. This would be a major breakthrough opening the door for quantum simulation experiments probing their properties in detail.

This work opens up several interesting research directions. Bipolarons can be considered as precursors of Cooper pairs in the limit of small impurity concentration, and a fundamental question is therefore how the nature of bipolarons change with increasing impurity concentration. This is analogous to the question concerning how the high $T_c$ superconducting phase emerges with increasing hole doping in the cuprates [29,30]. Another interesting problem concerns how the results presented in the present paper depend on temperature, and there are different theoretical and experimental results even for single polarons in continuum systems [15, 57–60]. Also, while the Fermi exclusion principle likely excludes bound states involving more fermionic impurities, a natural question is whether they exist for bosonic impurities. We have here concentrated on small values of $U_B/t_B$ where the bosons are deep in the superfluid regime, and an intriguing topic concerns the behaviour of the polarons and bipolarons for larger $U_B/t_B$ where the system undergoes a phase transition to a Mott insulating phase [43]. Moreover, since a range of interesting systems with strong correlations, non-trivial topology, and quantum phase transitions have been realised in optical lattices [37,38], an exciting new research direction is to use impurities as a probe of many-body physics [61–65]. Finally, we note that impurity dynamics in lattice systems is a topic of broad relevance also to condensed matter physics. Indeed, entirely new 2D Bose-Fermi lattice systems consisting of excitons mixed with electrons are now created in van der Waals bi-layers stacked with a relative twist angle [66]. In the limit of small electron concentration, this will realise lattice Bose polarons in a solid state setting. Excitons have moreover been used as impurities to detect Wigner crystals and Mott insulators in quantum materials [67,68].

# Acknowledgements

We acknowledge useful discussions with Zoe Yan. This work has been supported by the Danish National Research Foundation through the Center of Excellence "CCQ" (Grant agreement no.: DNRF156) (GMB, SD and AJ). G.A.D.-C. acknowledges a Consejo Nacional de Ciencia y Tecnología (CONACYT) scholarship and support of the Deutsche Forschungsgemeinschaft (DFG, German Research Foundation) under Germany's Excellence Strategy - EXC-2123 Quan-

tumFrontiers - 390837967. A. C. G. acknowledges financial support from Grant UNAM DGAPA PAPIIT No. IN108620 and PAPIIT No. IA101923.

## A Derivation of the Pair-Propagator in vacuum

As discussed in the main text, the vacuum pair-propagator $\Pi_v(\mathbf{0}, \omega)$ (see Eq.4) can be evaluated analytically. Here, we provide the main details of this derivation.

$$\Pi_v(\mathbf{0}, \omega) = \frac{1}{8\pi^2 t_B} \int_{-\pi}^{\pi} dk_x \int_{-\pi}^{\pi} dk_y \frac{1}{z' + 2\cos k_x + 2\cos k_y}, \tag{A.1}$$

where $z' = (\omega - 4t_B)/2t_B$. Performing the integration with respect to $k_y$, we obtain the following result for $|z'| > 4$:

$$\Pi_v(\mathbf{0}, \omega) = \frac{\mathrm{sgn}(z')}{2\pi t_B} \int_0^{\pi} \frac{dk_x}{\sqrt{(z' + 2\cos k_x)^2 - 4}}. \tag{A.2}$$

By expanding the quadratic term inside the root and introducing the new variable $s = \cos k_x$, it is straightforward to show that

$$\Pi_v(\mathbf{0}, \omega) = \frac{\mathrm{sgn}(z')}{4\pi t_B} \int_{-1}^{1} \frac{ds}{\sqrt{(1-s)(1+s)(\lambda_+ - s)(\lambda_- - s)}}, \tag{A.3}$$

where $\lambda_+ = 1 - z'/2$ and $\lambda_- = -1 - z'/2$. The resulting integral can be expressed in terms of the complete elliptic integral of the first kind $K(z')$ [69],

$$\Pi_v(\mathbf{0}, \omega) = \frac{1}{\pi z' t_B} K\left(\frac{4}{|z'|}\right), \tag{A.4}$$

which gives the result in Eq. 4 for $|z'| > 4$. To provide the expression for $|z'| < 4$, we employ the following analytic continuation of the elliptic function:

$$K\left(\frac{4}{|z'| + i\eta}\right) = \frac{|z'|}{4}\left(K\left(\frac{|z'|}{4}\right) - iK\left(\sqrt{1 - \frac{z'^2}{16}}\right)\right), \tag{A.5}$$

we then replace $z = z'/4$ to obtain Eq. 4 in the main text.

## B Two-body bound state

On a 2D square lattice, the two-body Schrödinger equation in real space with an on-site interaction is

$$\left[ -\sum_{\substack{i=1,2 \\ \sigma=x,y}} t_i \nabla^2_{i,\sigma} + U_{BI}\delta(\mathbf{r}_2 - \mathbf{r}_1) \right] \Psi(\mathbf{r}_1, \mathbf{r}_2) = E\Psi(\mathbf{r}_1, \mathbf{r}_2), \tag{B.1}$$

where $\nabla^2_{i,\sigma}\Psi(\mathbf{r}_i, \mathbf{r}_j) = \Psi(\mathbf{r}_i + \hat{\mathbf{e}}_\sigma, \mathbf{r}_j) - 2\Psi(\mathbf{r}_i, \mathbf{r}_j) + \Psi(\mathbf{r}_i - \hat{\mathbf{e}}_\sigma, \mathbf{r}_j)$ with $\hat{\mathbf{e}}_\sigma$ being the unit vector along the $\sigma$ direction. Expanding the wave function in terms of the plane wave $\Psi(\mathbf{r}_1, \mathbf{r}_2) = \sum_{\mathbf{k}_1, \mathbf{k}_2} \Phi(\mathbf{k}_1, \mathbf{k}_2) \exp(i\mathbf{k}_1\mathbf{r}_1 + i\mathbf{k}_2\mathbf{r}_2)$, we obtain for the kinetic energy

$$\sum_{\mathbf{k}_1, \mathbf{k}_2} \sum_{\substack{i=1,2 \\ \sigma=x,y}} [-2t_i(\cos k_{i\sigma} - 1)] \Phi(\mathbf{k}_1, \mathbf{k}_2) e^{i\mathbf{k}_1\mathbf{r}_1 + i\mathbf{k}_2\mathbf{r}_2}. \tag{B.2}$$

Since the interaction $U_{BI}\delta(\mathbf{r}_2 - \mathbf{r}_1)$ is only a function of the relative position $\mathbf{r}_2 - \mathbf{r}_1$, the total momentum is a conserved quantity. Then it is convenient to describe the system in terms of total momentum $\mathbf{P}$ and relative momentum $\mathbf{k}$. Introducing the center-of-mass and the relative position as [70]

$$\mathbf{R} = c\mathbf{r}_1 + (1-c)\mathbf{r}_2, \quad \text{and} \quad \mathbf{r} = \mathbf{r}_1 - \mathbf{r}_2, \tag{B.3}$$

where $c$ is a coefficient giving the weight of the position of particle 1 in the center of mass. Note that we are considering $t_{i,x} = t_{i,y}$ and consequently, $c_x = c_y = c$. Then the relation between $\mathbf{k}_1, \mathbf{k}_2$ and $\mathbf{P}, \mathbf{k}$ is

$$\begin{cases} \mathbf{P}c + \mathbf{k} = \mathbf{k}_1, \\ \mathbf{P}(1-c) - \mathbf{k} = \mathbf{k}_2. \end{cases} \tag{B.4}$$

We obtain the condition for $c$

$$\frac{t_1}{t_2} = \frac{\sin[P_\sigma(1-c)]}{\sin(P_\sigma c)}. \tag{B.5}$$

Then it is straightforward to derive that the kinetic energy in terms of $\mathbf{P}, \mathbf{k}$ for a specific $\mathbf{k}_1, \mathbf{k}_2$ is

$$K_{\mathbf{P},\mathbf{k}} = 2 \sum_{\sigma=x,y} \left[ E_{P_\sigma}(\cos k_\sigma - 1) + E_{P_\sigma} + t_1 + t_2 \right], \tag{B.6}$$

where $E_{P_\sigma} = -t_1 \cos(P_\sigma c) - t_2 \cos[P_\sigma(1-c)]$.

Since $\mathbf{P}$ is a conserved quantity, we fix $\mathbf{P}$ in the Schrödinger equation and project the equation to a specific relative momentum $\mathbf{k}$ state.

$$K_{\mathbf{P},\mathbf{k}}\varphi_{\mathbf{k}} + \sum_{\mathbf{k}'} U_{BI}\varphi_{\mathbf{k}'} = E\varphi_{\mathbf{k}}, \tag{B.7}$$

where for simplicity we denote the wavefunction as $\varphi_{\mathbf{k}}$. Therefore,

$$\varphi_{\mathbf{k}} = \frac{-\sum_{\mathbf{k}'} U_{BI}\varphi_{\mathbf{k}'}}{K_{\mathbf{P},\mathbf{k}} - E}. \tag{B.8}$$

After summing over $\mathbf{k}$ on both sides of the equation we obtain

$$\frac{1}{U_{BI}} = \sum_{\mathbf{k}} \frac{-1}{K_{\mathbf{P},\mathbf{k}} - E} = -\int d\epsilon \frac{N(\epsilon)}{\epsilon - E}, \tag{B.9}$$

where $\epsilon = K_{\mathbf{P},\mathbf{k}}$ and $N(\epsilon)$ is the corresponding density of states. For a fixed $\mathbf{P}$, $E_{P_\sigma}$ is a constant and therefore, $N(\epsilon)$ is qualitatively the same as that for a single particle on a rectangular lattice [71], which is finite near the boundary of the continuum. Therefore, for any value of $U_{BI} < 0/U_{BI} > 0$ one can always find a two-body bound state with $E$ below/above the continuum to satisfy Eq. B.9.

## C  Pair-Propagator in BEC

The impurity-boson pair-propagator in BEC is given by

$$\Pi(\mathbf{P}, i\Omega) = -\frac{1}{\beta M} \sum_{i\omega, \mathbf{k}\in BZ} G_{11}(i\omega, \mathbf{k}) G_{I0}(i\Omega - i\omega, \mathbf{P} - \mathbf{k}), \tag{C.10}$$

where $\beta = 1/k_B T$ with $k_B$ being the Boltzmann constant and $T$ the temperature. $G_{I0}(i\omega, \mathbf{k}) = 1/(i\omega - \epsilon_{I\mathbf{k}})$ is the Green's function for bare impurity and the BEC normal Green's function is

$$G_{11}(i\omega, \mathbf{k}) = \frac{u_{\mathbf{k}}^2}{i\omega - E_{\mathbf{k}}} - \frac{v_{\mathbf{k}}^2}{i\omega + E_{\mathbf{k}}}, \tag{C.11}$$

with $v_{\mathbf{k}}^2 = u_{\mathbf{k}}^2 - 1$. It is straightforward to derive

$$
\begin{aligned}
\frac{1}{\beta}\sum_{i\omega} G_{11}(i\omega, \mathbf{k})\, G_{I0}(i\Omega - i\omega, \mathbf{P} - \mathbf{k}) ={}& \frac{u_{\mathbf{k}}^2 n_B(E_{\mathbf{k}})}{-i\Omega + E_{\mathbf{k}} + \epsilon_{I\mathbf{P}-\mathbf{k}}} - \frac{v_{\mathbf{k}}^2 n_B(-E_{\mathbf{k}})}{-i\Omega - E_{\mathbf{k}} + \epsilon_{I\mathbf{P}-\mathbf{k}}} \\
&+ n_B(i\Omega - \epsilon_{I\mathbf{P}-\mathbf{k}})\left(\frac{u_{\mathbf{k}}^2}{i\Omega - \epsilon_{I\mathbf{P}-\mathbf{k}} - E_{\mathbf{k}}} - \frac{v_{\mathbf{k}}^2}{i\Omega - \epsilon_{I\mathbf{P}-\mathbf{k}} + E_{\mathbf{k}}}\right),
\end{aligned}
\tag{C.12}
$$

where $n_B(x) = 1/(\exp(\beta x) - 1)$ is the bosonic distribution function. For bosonic impurities $i\Omega = i2n\pi/\beta$ ( with $n$ an integer), $n_B(i\Omega - \epsilon_{I\mathbf{P}-\mathbf{k}}) = n_B(-\epsilon_{I\mathbf{P}-\mathbf{k}})$. For a single impurity, the distribution function of the impurity vanishes, thus $n_B(i\Omega - \epsilon_{I\mathbf{P}-\mathbf{k}}) = -1$ and at $T = 0$, the pair-propagator is

$$\Pi(\mathbf{P}, i\Omega) = \frac{1}{M}\sum_{\mathbf{k}} \frac{u_{\mathbf{k}}^2}{i\Omega - \epsilon_{I\mathbf{P}-\mathbf{k}} - E_{\mathbf{k}}}. \tag{C.13}$$

For fermionic impurities the Matsubara frequencies $i\Omega = i(2n+1)\pi/\beta$, which yields $n_B(i\Omega - \epsilon_{I\mathbf{P}-\mathbf{k}}) = -f_F(-\epsilon_{I\mathbf{P}-\mathbf{k}})$ with the Fermi distribution function $f_F(x) = 1/(\exp(\beta x) + 1)$. Considering the limit of a single impurity, one still has $n_B(i\Omega - \epsilon_{I\mathbf{P}-\mathbf{k}}) = -1$ and at $T = 0$, Eq. C.13 . Therefore, one ends up with Eq. 7 for the pair propagator in the main text.

## D  Spatial Correlations, Wavefunction and $\mathcal{T}$-matrix

*Spatial correlations.-* Let us provide further details on the spatial properties of the polaron, such features are obtained from a many-body polaron wavefunction equivalent to the $\mathcal{T}$-matrix formalism. Let us start discussing the spatial correlations given by Eq. 10 in the main text. In Eq. 10, $\hat{n}_B(\mathbf{i}) = \hat{b}_{\mathbf{i}}^{\dagger}\hat{b}_{\mathbf{i}}$, which we conveniently write in momentum space to account for the condensed atoms forming the BEC, thus, we have

$$
\begin{aligned}
\hat{n}_B(\mathbf{i}) ={}& n_0 + \frac{\sqrt{n_0}}{\sqrt{M}}\sum_{\mathbf{q}\neq 0}\left(e^{-i\mathbf{q}\cdot\mathbf{i}}\hat{b}_{\mathbf{q}}^{\dagger} + e^{i\mathbf{q}\cdot\mathbf{i}}\hat{b}_{\mathbf{q}}\right) + \frac{1}{M}\sum_{\substack{\mathbf{k}\neq 0 \\ \mathbf{q}\neq\mathbf{k}}}e^{-i\mathbf{q}\cdot\mathbf{i}}\hat{b}_{\mathbf{k}}^{\dagger}\hat{b}_{\mathbf{k}-\mathbf{q}} \\
={}& n_0 + n_{\text{ex}} + \frac{\sqrt{n_0}}{\sqrt{M}}\sum_{\mathbf{q}}(u_{\mathbf{q}} + v_{\mathbf{q}})\left(e^{-i\mathbf{q}\cdot\mathbf{i}}\hat{\beta}_{\mathbf{q}}^{\dagger} + e^{i\mathbf{q}\cdot\mathbf{i}}\hat{\beta}_{\mathbf{q}}\right) \\
&+ \frac{1}{M}\sum_{\mathbf{k},\mathbf{q}}(e^{-i\mathbf{q}\cdot\mathbf{i}}\hat{\beta}_{\mathbf{k}}^{\dagger}\hat{\beta}_{\mathbf{k}-\mathbf{q}}u_{\mathbf{k}}u_{\mathbf{k}-\mathbf{q}} + e^{i\mathbf{q}\cdot\mathbf{i}}\hat{\beta}_{\mathbf{k}-\mathbf{q}}^{\dagger}\hat{\beta}_{\mathbf{k}}v_{\mathbf{k}}v_{\mathbf{k}-\mathbf{q}}) \\
&+ \frac{1}{M}\sum_{\mathbf{k},\mathbf{q}}e^{-i\mathbf{q}\cdot\mathbf{i}}(\hat{\beta}_{\mathbf{k}}^{\dagger}\hat{\beta}_{\mathbf{q}-\mathbf{k}}^{\dagger}u_{\mathbf{k}}v_{\mathbf{q}-\mathbf{k}} + \hat{\beta}_{-\mathbf{k}}\hat{\beta}_{\mathbf{k}-\mathbf{q}}v_{-\mathbf{k}}u_{\mathbf{k}-\mathbf{q}}),
\end{aligned}
\tag{D.1}
$$

written in terms of the standard Bogoliubov operators

$$\hat{b}_{\mathbf{k}} = u_{\mathbf{k}}\hat{\beta}_{\mathbf{k}} + v_{-\mathbf{k}}\hat{\beta}_{-\mathbf{k}}^{\dagger}, \tag{D.2 a}$$

$$\hat{b}_{\mathbf{k}}^{\dagger} = u_{\mathbf{k}}\hat{\beta}_{\mathbf{k}}^{\dagger} + v_{-\mathbf{k}}\hat{\beta}_{-\mathbf{k}}. \tag{D.2 b}$$

$n_{\text{ex}} = \sum_{\mathbf{k}\neq 0} v_{\mathbf{k}}^2/M$ is the density of the depletion of the ground state BEC due to finite $U_B$. To evaluate the correlation function we employ the many-body polaron wavefunction defined by Eq. 9 in the main text and take the approximation

$n = \langle \Psi_P | \hat{n}_B(\mathbf{i}) | \Psi_P \rangle = \langle BEC | \hat{n}_B(\mathbf{i}) | BEC \rangle = n_0 + n_{\text{ex}} = n_0$. After some algebra we obtain

$$
\begin{aligned}
\langle \Psi_P | [\hat{n}_B(\mathbf{i}) - n] \hat{n}_I(\mathbf{0}) | \Psi_P \rangle = & \frac{\sqrt{n}}{M^{3/2}} \sum_{\mathbf{p}} (u_{-\mathbf{p}} + v_{-\mathbf{p}}) \left( \phi_0^* \psi_{\mathbf{p}} e^{-i\mathbf{p} \cdot \mathbf{i}} + \phi_0 \psi_{\mathbf{p}}^* e^{i\mathbf{p} \cdot \mathbf{i}} \right) \\
& + \frac{1}{M^2} \sum_{\mathbf{p}, \mathbf{p}'} e^{-i(\mathbf{p} - \mathbf{p}') \cdot \mathbf{i}} \psi_{\mathbf{p}'}^* \psi_{\mathbf{p}} (u_{-\mathbf{p}} u_{-\mathbf{p}'} + v_{\mathbf{p}} v_{\mathbf{p}'}).
\end{aligned}
\tag{D.3}
$$

This expression depends on the properties of the BEC as well as the parameters of the many-body wave function $\phi_0$ and $\psi_{\mathbf{k}}$. We make use of the properties $u_{\mathbf{k}} = u_{-\mathbf{k}}$, and $v_{\mathbf{k}} = v_{-\mathbf{k}}$, as well as the wavefunction follows $\psi_{\mathbf{k}} = \psi_{-\mathbf{k}}$, as we will explicitly show.

*Many-body wavefunction.-* To determine the parameters of Eq. 9 we employ a variational approach which as we will illustrate is equivalent to the field theory employed with the self-energy as in Eq. 6. We start from the energy functional for $\omega_0$

$$
\begin{aligned}
\omega_0 & \left( |\phi_0|^2 + \sum_{\mathbf{k}} |\psi_{\mathbf{k}}|^2 \right) = \sum_{\mathbf{k}} (\epsilon_{I\mathbf{k}} + E_{\mathbf{k}}) |\psi_{\mathbf{k}}|^2 + \epsilon_{I\mathbf{0}} |\phi_0|^2 + n U_{BI} |\phi_0|^2 \\
& + \frac{U_{BI} \sqrt{n_0}}{\sqrt{M}} \sum_{\mathbf{k}} (u_{\mathbf{k}} + v_{\mathbf{k}}) (\phi_0 \psi_{\mathbf{k}}^* + \phi_0^* \psi_{\mathbf{k}}) + \frac{U_{BI}}{M} \sum_{\mathbf{p}, \mathbf{p}'} (u_{-\mathbf{p}} u_{-\mathbf{p}'} + v_{\mathbf{p}'} v_{\mathbf{p}}) \psi_{\mathbf{p}'}^* \psi_{\mathbf{p}},
\end{aligned}
\tag{D.4}
$$

where $\phi_0$ and $\phi_0^*$ as well as $\psi_{\mathbf{k}}$ and $\psi_{\mathbf{k}}^*$ are treated as independent parameters. Then, by minimizing the energy with respect to $\phi_0^*$ and $\psi_{\mathbf{k}}^*$ we obtain the following set of equations

$$
0 = \frac{U_{BI} \sqrt{n}}{\sqrt{M}} \sum_{\mathbf{k}} u_{\mathbf{k}} \psi_{\mathbf{k}} - (\omega_0 - \epsilon_{I\mathbf{0}} - n U_{BI}) \phi_0,
\tag{D.5 a}
$$

$$
0 = (\epsilon_{I\mathbf{k}} + E_{\mathbf{k}} - \omega_0) \psi_{\mathbf{k}} + \frac{U_{BI} \sqrt{n}}{\sqrt{M}} u_{\mathbf{k}} \phi_0 + u_{\mathbf{k}} \frac{U_{BI}}{M} \sum_{\mathbf{p}} u_{\mathbf{p}} \psi_{\mathbf{p}},
\tag{D.5 b}
$$

where for consistency with our Green's function formalism we only retain the $u_{\mathbf{k}}$ coherence factors and take $n_0 = n$.

To determine the amplitude of the wavefunction in terms of the quasiparticle properties we have from the Eq. D.5 a

$$
\sum_{\mathbf{k}} u_{\mathbf{k}} \psi_{\mathbf{k}} = \frac{\sqrt{M}}{U_{BI} \sqrt{n}} (\omega_0 - \epsilon_{I\mathbf{0}} - n U_{BI}) \phi_0,
\tag{D.6}
$$

which can be replaced into the Eq. D.5 b. Thus, if the polaron energy and its quasiparticle residue are known, then we can directly determine the many-body wave function with

$$
\psi_{\mathbf{k}} = -\frac{\omega_0 - \epsilon_{I\mathbf{0}}}{\epsilon_{I\mathbf{k}} + E_{\mathbf{k}} - \omega_0} \frac{u_{\mathbf{k}} \sqrt{Z_0}}{\sqrt{Mn}},
\tag{D.7}
$$

where the amplitude $\phi_0 = \sqrt{Z_0}$ has been applied.

*Wavefunction and the $\mathcal{T}$-matrix.-* To obtain the polaron wavefunction we employ the quasiparticle properties from the $\mathcal{T}$-matrix formalism, this is justified by the equivalence between these two approaches as we will briefly discuss. To prove the equivalence between these approaches, let us introduce the following parameter,

$$
\chi = \phi_0 + \sum_{\mathbf{k}} u_{\mathbf{k}} \psi_{\mathbf{k}}.
\tag{D.8}
$$

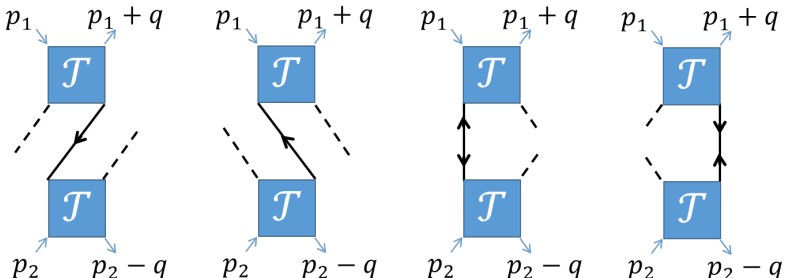

Figure 9: The diagram for the induced interaction between two impurities due to the exchange of the Bogoliubov mode. The solid black lines are the normal and anomalous Bogoliubov Green's functions for the bosons, dashed lines correspond to condensate particles. The boson-impurity interaction is represented by the blue boxes and is taken within the ladder approximation.

Writing $\phi_0$ in terms of $\chi$ using Eq. D.5 a we obtain

$$\phi_0 = U_{BI}\sqrt{\frac{n}{M}} \frac{\chi}{\omega_0 - \epsilon_{I0} - nU_{BI} + U_{BI}\sqrt{\frac{n}{M}}}. \tag{D.9}$$

Further replace this expression into Eq. D.5 b to write $\psi_{\mathbf{k}}$ in terms of $\chi$

$$\psi_{\mathbf{k}} = \frac{u_{\mathbf{k}}}{\omega_0 - \epsilon_{I\mathbf{k}} - E_{\mathbf{k}}} \frac{U_{BI}}{M} \frac{(\omega_0 - \epsilon_{I0})\chi}{\omega_0 - \epsilon_{I0} - nU_{BI} + U_{BI}\sqrt{\frac{n}{M}}}. \tag{D.10}$$

Now, we replace Eq. D.9 and Eq. D.10 in the expression for $\chi$ in Eq. D.8

$$\chi = \frac{\chi}{\omega_0 - \epsilon_{I0} - nU_{BI} + U_{BI}\sqrt{\frac{n}{M}}} \left[ U_{BI}\sqrt{\frac{n}{M}} + \frac{U_{BI}}{M} \sum_{\mathbf{k}} \frac{u_{\mathbf{k}}^2(\omega_0 - \epsilon_{I0})}{\omega_0 - \epsilon_{I\mathbf{k}} - E_{\mathbf{k}}} \right]. \tag{D.11}$$

After some algebra we arrive at a self-consistent equation for $\omega_0$

$$\omega_0 - \epsilon_{I0} = \frac{nU_{BI}}{1 - \frac{U_{BI}}{M}\sum_{\mathbf{k}}\frac{u_{\mathbf{k}}^2}{\omega_0 - \epsilon_{I\mathbf{k}} - E_{\mathbf{k}}}} = n\mathcal{T}(\omega_0), \tag{D.12}$$

thus the quasiparticle energy of the polaron is given by

$$\omega_0 = \epsilon_{I0} + n\mathcal{T}(\omega_0), \tag{D.13}$$

which is precisely the equation for the polaron energy employed within the Green's function formalism.

# E  Induced interaction between polarons

The induced interaction between two impurities mediated by a BEC is given by the diagram shown in Figure. 9. The analytical expression is

$$V_{\text{ind}}^I(p_1, p_2, q) = n_0[\mathcal{T}(p_1)G_{11}(-q)\mathcal{T}(p_2 - q) + \mathcal{T}(p_1 + q)G_{11}(q)\mathcal{T}(p_2)$$
$$+ \mathcal{T}(p_1 + q)G_{12}(q)\mathcal{T}(p_2 - q) + \mathcal{T}(p_1)G_{21}(q)\mathcal{T}(p_2)], \tag{E.1}$$

where $p_1, p_2, q$ are four momentums and the anomalous Green's functions of BEC

$$G_{12}(i\omega, \mathbf{k}) = G_{21}(i\omega, \mathbf{k}) = u_{\mathbf{k}}v_{\mathbf{k}}\left(\frac{1}{i\omega - E_{\mathbf{k}}} - \frac{1}{i\omega + E_{\mathbf{k}}}\right), \tag{E.2}$$

with $u_{\mathbf{k}} v_{\mathbf{k}} = -n_0 U_B / 2 E_{\mathbf{k}}$. Suppose $\tilde{E}_B$ is the binding energy of the produced bipolaron and $m^*$ the effective mass of the polaron. If $\sqrt{|\tilde{E}_B|/m^*}$ is far less than the sound velocity of the BEC, the time consumed by the density wave in BEC to transport the induced interaction between polarons can be ignored. Therefore, $V_{\text{ind}}^I(p_1, p_2, q)$ is approximately a frequency transfer $iq_0$ independent function. In principal, we can take any value for $iq_0$ in $V_{\text{ind}}^I$ and in our calculation, we take $iq_0 = 0$. Since the variations of the energies of polarons during scattering should be of order of $\tilde{E}_B$, one can approximately treat the energy components of $p_1$ and $p_2$ in $V_{\text{ind}}^I$ as appropriate constants if $\tilde{E}_B$ is far less than the typical energy of the Bogoliubov excitation. As a result, the induced interaction between a pair of polarons with total momentum being 0 is given by

$$
\begin{aligned}
V_{\text{ind}}\left(\mathbf{k}, \mathbf{k}'\right) = Z_{\mathbf{k}}^2 n_0 [\, & \mathcal{T}\left(\omega, \mathbf{k}\right) G_{11}\left(0, \mathbf{k} - \mathbf{k}'\right) \mathcal{T}\left(\omega, -\mathbf{k}'\right) + \mathcal{T}\left(\omega, \mathbf{k}'\right) G_{11}\left(0, \mathbf{k}' - \mathbf{k}\right) \mathcal{T}\left(\omega, -\mathbf{k}\right) \\
& + \mathcal{T}\left(\omega, \mathbf{k}'\right) G_{12}\left(0, \mathbf{k}' - \mathbf{k}\right) \mathcal{T}\left(\omega, -\mathbf{k}'\right) + \mathcal{T}\left(\omega, \mathbf{k}\right) G_{21}\left(0, \mathbf{k}' - \mathbf{k}\right) \mathcal{T}\left(\omega, -\mathbf{k}\right) ],
\end{aligned}
\tag{E.3}
$$

where $\mathbf{k}$ and $\mathbf{k}'$ are respectively the relative momentums before and after scattering. Considering

$$
G_{11}\left(i\omega, \mathbf{k}\right) = G_{11}\left(i\omega, -\mathbf{k}\right), \tag{E.4 a}
$$

$$
\mathcal{T}\left(i\omega, \mathbf{k}\right) = \mathcal{T}\left(i\omega, -\mathbf{k}\right), \tag{E.4 b}
$$

one has the Eq. 15 in the main text.

Now $V(p_1, p_2; q_1)$ can be taken out from the frequency summation in Eq. 12. If the pole expansion for impurity Green's function $G_I(i\omega, \mathbf{k}) \simeq Z_{\mathbf{k}} / (i\omega - \omega_{\mathbf{k}})$ is taken, this summation can be calculated analytically,

$$
\frac{1}{\beta} \sum_{iq_0} G_I\left(iq_0, \mathbf{k}\right) G_I\left(i\Omega - iq_0, -\mathbf{k}\right) = -\frac{Z_{\mathbf{k}}^2}{i\Omega - 2\omega_{\mathbf{k}}}, \tag{E.5}
$$

which only depends on the total frequency $i\Omega$ of the two polarons. As a result, the Bethe-Salpeter equation reduces to be the Lippmann-Schwinger equation

$$
\Gamma_{\text{eff}}\left(\mathbf{k}, \mathbf{k}', i\Omega\right) = V_{\text{qp}}\left(\mathbf{k}, \mathbf{k}'\right) + \frac{1}{M} \sum_{\mathbf{q}_1} \frac{V_{\text{qp}}\left(\mathbf{k}, \mathbf{k} + \mathbf{q}_1\right)}{i\Omega - 2\omega_{\mathbf{k}+\mathbf{q}_1}} \Gamma_{\text{eff}}\left(\mathbf{k} + \mathbf{q}_1, \mathbf{k}', i\Omega\right), \tag{E.6}
$$

where we have changed the notation so that the three arguments $\mathbf{k}, \mathbf{k}'$ and $i\Omega$ in $\Gamma_{\text{eff}}$ respectively represent the relative momentum of incoming and outgoing polarons as well as the total frequency of the two polarons. Eq. E.6 is equivalent to the Schrödinger equation Eq. 13.

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
