# Peer review of "Polarons and bipolarons in a two-dimensional square lattice"

_SciPost Physics, doi:SciPost Phys. 14, 143 (2023)_

## Round 1 · Referee Report · Anonymous (Referee 1) · 2022-12-22

Report

The work by Ding et al. studies the dynamics of impurities immersed in a weakly interacting Bose gas on a two-dimensional square lattice. First the spectral function of a single impurity in the Bose gas is studied. To this end the self-energy is computed with the T-matrix approximation. Both attractively and repulsively bound states (=polarons) are found. Particular focus is then put on the study of effective mutual interactions between impurities, arising from the exchange of bosons in the BEC. Due to the effective attractive interaction a bound state of two polarons forms, known as bipolaron.

The results of the paper are relatively clearly presented and contain new aspects. Given the advancements of quantum simulators, the topic is also timely. A couple of questions and comments arose when reading the paper:

1) Can the authors analyze or at least comment on finite temperature effects? Would the bound state of the impurities be modified in this case?
2) Can the authors comment on whether more complex multi-impurity bound states can arise in this setting? Given the simplifications, which have been used to reduce the Bethe-Salpeter equation to a easily-tractable equation, it may be possible to consider also higher order bound states.
3) What are the expectations when the bosons are driven near a Mott transition or even into an insulator?
4) A minor comment: In the beginning of Chapter 6 the question of whether a bound state is formed for the effective attractive interactions has been asked. Is there a reason not to expect the bound state to form, given the system is in 2D and the potential is attractive? If this is the expectation, then I'd suggest to reformulate.

I believe, that the paper could benefit from a discussion of the above-mentioned comments, as some extra focus is put on the central aspects of the lattice. Once taken into account, the papers can be accepted in Sci Post.

---

## Round 1 · Referee Report · Anonymous (Referee 2) · 2023-1-10

Strengths

1 - The considered topic is interesting
2 - The paper is written well

Weaknesses

1 - some figures can be improved

Report

The Authors perform an analytical study of the problem of impurities in a 2D square lattice. The polaron problem is related to the current experiments and the obtained results are relevant in that context. I find the article to be well-written and I recommend its acceptance once my comments are addressed.

Requested changes

The following comments should be addressed 1 - unit filling $n_0 = 1$ is used "for concreteness". It would be useful to have a comment if important differences are expected for different values of $n_0$ 2 - Induced interactions: explain better the region of applicability of the results 3 - Induced interactions: is it possible to comment on what is the behavior of the long-range tail (power law, exponential, etc)? 4- I advise adding Fig 3b with $k_y=0$ data 5 - Make figures readable in black and white version (Fig. 3, etc) 6- "Here and in the rest of the paper we take the value $U_B/t_B = 0.07$." Provide motivation for using this specific value. 7- Below Eq. (4), "We have defined $z =$ ...". Explain the physical meaning of $z$. 8- In the used notation indices "Bk", "Ik" in the energy spectrum should be deciphered as index + argument, which might be confusing. Instead, double arguments are given with parenthesis $(k,\omega)$. It would be less confusing use $(k)$ as an argument and B/I as an index. 9- "... scattering of an impurity atom and a boson ...". I think "atom" can be omitted in similar phrases, or atom should be added also to "boson", as the scattering occurs between two atoms, instead one is called an atom and the other is not. 10- "we assume that the interaction $U_B$ is weak so that it is accurately described by Bogoliubov theory". What should be accurately described by the Bogoliubov theory? Interaction? 11- "states with center of mass momentum zero", rephrase. 12- "We denote ... as the upper polaron", I would advise to emphasize the name "{\em upper polaron}" 13- "Equation (8) shows ... residue" -> "Equation (8a)" 14- The placement of the figures is weird, the first one is inside the text while the rest of the figures are at the end of the paper. Arrange figures properly. 15 - Introduction, "BECs" abbreviation is not introduced in the text of the article 16- below Eq .(1), "Here, $\hat b_i$..." -> "Here, operators $\hat b_i$..." 17- In the Abstract, "... the attractive nature of the effective interactions between two polarons combined with the two-dimensionality ...". It is a general feature of the second-order perturbative theories to provide a negative correction so that attractive interactions appear in all dimensionalities. Is the low dimensionality a keypoint here? Or should it be necessarily 2D? 18- Fig. 2, I advise to a add a vertical black line at $x=0$ position (no additional caption is needed) 19- Fig. 2, the horizontal dashed line seems to correspond to 1/2 and it is not clear if this is a coincidence as typically 1/2 is not a special value for the residue 20- Fig. 4, Figures are too small, 0 and 1 values are not shown. The projection angle in (b) panel is not optimal. Alternatively, top view as in Fig. 5 can be tried. Try to improve the figures.

---

## Round 3 · Referee Report · Anonymous (Referee 3) · 2023-2-13

Report

I find the article to be well-written and that it provides new and interesting results on the problem of polarons in two-dimensional geometry. It meets the general acceptance criteria for being published in SciPost.

The Authors did a good job in answering the raised comments. From that, I am satisfied with their answers and recommend the publication.

As well, I provide a few additional comments which might be optionally addressed. The numbering follows the comments from the previous report.

1. Using $n_0=1$ in lattice models might seem to be a special, commensurate value, and result in significantly different properties as compared to non-commensurate values. It is appropriate to verify and comment that, indeed, changing $n_0$ slightly does not lead to significant changes.

7. Is It possible to argue why $8t_B$ is a physically relevant scale to compare with?

8. I do not see what is the problem with showing the argument explicitly in round brackets. On the other hand, that is the Authors' choice of using an appropriate notation and avoiding confusion.

  • validity: -
  • significance: -
  • originality: -
  • clarity: -
  • formatting: -
  • grammar: -

Author:  Shanshan Ding  on 2023-04-17  [id 3594]

(in reply to Report 1 on 2023-02-13)

We thank the referee for the second round of review. Below, we address each point raised in his/her report.

>1. Using n0=1 in lattice models might seem to be a special, commensurate value, and result in significantly different properties as compared to non-commensurate values. It is appropriate to verify and comment that, indeed, changing n0 slightly does not lead to significant changes.

Our response:
We agree that $n_0=1$ is indeed a special case for the Mott insulator. However, for the superfluid state that we consider, $n_0=1$ is not special because all particles in the lattice are in a coherent state. Hence, changing $n_0$ slightly will not lead to significant changes.

>7. Is It possible to argue why 8tB is a physically relevant scale to compare with?

Our response:
$8t_B$ is the bandwidth of the single particle band and therefore a relevant energy scale.

In response to the referee, we have added “the bandwidth of a single particle” before “$8t_B$” in the manuscript.

>8. I do not see what is the problem with showing the argument explicitly in round brackets. On the other hand, that is the Authors' choice of using an appropriate notation and avoiding confusion.

Our response:
Notation is largely a matter of taste and we agree that using brackets has its advantages. However, we respectfully prefer using subscripts since this avoids clutter and nested brackets.

---

## Round 3 · Referee Report · Anonymous (Referee 4) · 2023-3-9

Report

The paper can be accepted in SciPost physics.

---

## Round 3 · Author Response

Dear Editor,

We thank the editor and the referees for the careful review of our paper and appreciate that both referees recommend publication in *SciPost Physics*, once we have taken their comments into consideration. The feedback from both referees has inspired us to improve the manuscript, and our main revisions can be summarized as:

1.We have added a new panel in Fig. 3 plotting the energy of the polaron as a function of $k_x$ for $k_y=0$.

2.We have made minor modifications to some of the figures to make them more readable in black and white.

3.We have made sure that the figures appear at appropriate places in the text.

4.We have clarified the conditions for the approximations leading to Eq. (15).

5.We give an expression for the long-range tail of the induced interaction for weak coupling.

6.We have extended the *Conclusions and Outlook* section to discuss the interesting open problems raised by the referees that are outside the scope of the present paper.

7.The text has been clarified in a few places.

Below, we give a detailed reply to all comments of the two referees and describe the corresponding changes to the manuscript. We hereby resubmit our manuscript to *SciPost Physics*.

Sincerely,

Shanshan Ding, G. A. Domínguez-Castro, Aleksi Julku, Arturo Camacho-Guardian and Georg M. Bruun

**Reply to Report 1.**

We thank the referee for the very careful review and feedback that has helped us to improve our manuscript. Below, we address each point raised by the referee in detail.

>*1) Can the authors analyze or at least comment on finite temperature effects? Would the bound state of the impurities be modified in this case?*

The referee raises an interesting but challenging point as the effects of temperature enter in several places. The dominant contribution to induced interaction for low temperatures is proportional to the condensate density $n_0$ as can be seen explicitly in Eq. (15). This will lead to a decrease in the strength of the interaction as the condensate gets depleted with increasing temperature. There are however other terms corresponding giving the contribution to the induced interaction mediated by thermal bosons, that may partially compensate for this. Also, a decrease in the condensate density will lead to a decrease in the speed of sound in the BEC, which also affects the induced interaction.

While a detailed analysis of these effects of a non-zero temperature is intriguing and non-trivial, it is outside the scope of the present paper. Instead, we mention this is an interesting future research problem in the *Conclusions and Outlook* section.

>*2) Can the authors comment on whether more complex multi-impurity bound states can arise in this setting? Given the simplifications, which have been used to reduce the Bethe-Salpeter equation to a easily-tractable equation, it may be possible to consider also higher order bound states.*

For bosonic impurities, bound states involving more impurities may indeed exist, at least in a vacuum. It is however unclear if such states are stable in a many-body environment as they might be destroyed by scattering on bosons in the BEC. We do on the other hand not expect such multi-impurity states to exist for fermionic impurities due to the Pauli repulsion.

We agree that the presence of multi-impurity bound states is an interesting problem but it is also quite challenging – in particular in a many-body environment. In general, one would have to analyze the multi-body scattering matrix, which is exceedingly complex. Perhaps one can in certain regimes reduce this to a sum of two-body scattering processes and thus use some of the results in the present paper, but it requires a careful analysis to investigate when such approximations are reliable. We have therefore not included such an analysis in the present paper. It is instead mentioned as an interesting problem in the *Conclusions and Outlook* section.

>*3) What are the expectations when the bosons are driven near a Mott transition or even into an insulator?*

Deep in the Mott insulating phase, the spectrum is gapped and the bosons are essentially incompressible. This will make the induced interaction very weak, and we do not expect that it will support bound states. Close to the superfluid-insulating transition on the other hand, a collective mode in general becomes gapless, which may make a large contribution to the induced interaction.

In response to the referee, we now briefly discuss the interesting question of bi-polarons in the insulator-superfluid transition region in the *Conclusions and Outlook* section.

>*4) A minor comment: In the beginning of Chapter 6 the question of whether a bound state is formed for the effective attractive interactions has been asked. Is there a reason not to expect the bound state to form, given the system is in 2D and the potential is attractive? If this is the expectation, then I'd suggest to reformulate.*

The referee raises an interesting point. It is known that for a local interaction $V(r)$ in a 2D homogeneous system, the condition for the existence of a bound state is $\int V(r)rdr<0$, see for example Simon, *Annals of Physics* (1976) or *Quantum mechanics* by Landau & Lifshitz. The case analyzed in the present paper is however different. First, we are considering a lattice system with a single particle dispersion different from $p^2/2m$. Second, the induced interaction appearing in the Schrödinger equation is non-local as discussed below Eq. (13). It is therefore not obvious how to generalize the result stated above to the case at hand. We now explain this more carefully in the beginning of Sec. 6 and cite the book by Landau & Lifshitz.

**Reply to Report 2.**

We thank the referee for the very careful review that has helped us to improve our manuscript. Below, we address each point raised by the referee in detail.

>*1 - unit filling n0=1 is used "for concreteness". It would be useful to have a comment if important differences are expected for different values of n0*

We thank referee for pointing this out. Our analytical expressions, see e.g. Eq. (6), (8) and (15), are valid for a general value of $n_0$ as long as the BEC is weakly interacting so that Bogoliubov theory applies. We only use the specific value $n_0=1$ for the numerical calculations and we expect that the results presented will vary smoothly with $n_0$.

We have now moved the remark concerning taking $n_0=1$ down to where we first present our numerical results to avoid confusion.

>*2 - Induced interactions: explain better the region of applicability of the results.*

We could indeed have explained this point more clearly. Our results for the induced interaction and bi-polarons are valid under the following conditions:

1.The single polaron is well-defined so that one can apply the pole expansion for the Green’s functions.

2.The binding energy of the bipolaron is sufficiently small so that retardation effects (frequency dependence) of the induced interaction can be ignored, and the energy of the polarons during scattering can be approximated as being constant.

We have in response to the referee emphasized these points even more in the new version of the manuscript before Eq. (13).

>*3 - Induced interactions: is it possible to comment on what is the behavior of the long-range tail (power law, exponential, etc)?*

This is a good point. For the general case, the induced interaction given by Eq. (15) is non-local depending on both the incoming and outgoing relative momentum, and its spatial dependence is therefore non-trivial. For weak impurity-boson coupling on the other hand, the induced interaction simplifies to Eq. (16), which only depends on the momentum transfer. Fourier transforming assuming a quadratic single particle dispersion at small momenta gives the long-range form $\propto\exp(-\sqrt{2n_0U_B/t_B}r)/\sqrt{r}$.

We now discuss this long range form in the weak coupling limit below Eq. (16).

>*4- I advise adding Fig 3b with ky=0 data.*

The plot along the $k_x$-axis for $k_y=0$ is added in the new version as Fig. 3b.

>*5 - Make figures readable in black and white version (Fig. 3, etc).*

We have made the figures readable in black and white as much as possible by modifying the descriptions of Figs. 3, 7, and 8, and by changing the lines in Fig. 5 (d).

>*6- "Here and in the rest of the paper we take the value UB/tB=0.07." Provide motivation for using this specific value.*

We choose such a small value to ensure that the bosons are far away from the Mott regime. As shown in the experiment described in PRL **105**, 110401 (2010), this value is experimentally realistic. We now explain this in the manuscript.

>*7- Below Eq. (4), "We have defined z= ...". Explain the physical meaning of z.*

The variable $z$ is the two-body energy measured in units of $8t_B$. We explain this immediately after its definition in the manuscript.

>*8- In the used notation indices "Bk", "Ik" in the energy spectrum should be deciphered as index + argument, which might be confusing. Instead, double arguments are given with parenthesis (k,ω). It would be less confusing use (k) as an argument and B/I as an index.*

We agree with the referee that this notation might be a bit confusing. However, we respectfully prefer to keep it since we think the alternative is worse, as it will result in many nested brackets in equations such as that just below Eq. (7). Our notation is also consistent with using $u_k, E_k, \cdots$.

>*9- "... scattering of an impurity atom and a boson ...". I think "atom" can be omitted in similar phrases, or atom should be added also to "boson", as the scattering occurs between two atoms, instead one is called an atom and the other is not.*

The referee is correct, and we have now removed the word “atom” in such phrases.

>*10- "we assume that the interaction UB is weak so that it is accurately described by Bogoliubov theory". What should be accurately described by the Bogoliubov theory? Interaction?*

We thank the referee for the careful reading. We have replaced “it” by “the BEC” in the new version of the manuscript.

>*11- "states with center of mass momentum zero", rephrase.*

The sentence now reads “states with zero center of mass momentum”.

>*12- "We denote ... as the upper polaron", I would advise to emphasize the name "{\em upper polaron}".*

We agree with the suggestion of the referee and have revised the manuscript accordingly.

>*13- "Equation (8) shows ... residue" -> "Equation (8a)".*

This has now been fixed.

>*14- The placement of the figures is weird, the first one is inside the text while the rest of the figures are at the end of the paper. Arrange figures properly.*

The figures are now placed properly.

>*15 - Introduction, "BECs" abbreviation is not introduced in the text of the article.*

The abbreviation “BEC” has now been defined when it is first used.

>*16- below Eq .(1), "Here, $\hat{b}_{\mathbf i}$..." -> "Here, operators $\hat{b}_{\mathbf i}$...".*

We follow the suggestion of the referee and the manuscript has been revised accordingly.

>*17- In the Abstract, "... the attractive nature of the effective interactions between two polarons combined with the two-dimensionality ...". It is a general feature of the second-order perturbative theories to provide a negative correction so that attractive interactions appear in all dimensionalities. Is the low dimensionality a keypoint here? Or should it be necessarily 2D?*

The referee is completely correct that it is a general feature of second order perturbation theory to predict an attractive induced interaction. However, it is not obvious that this is enough to lead to bound states. In general, bound states in 2D are easier to form than in 3D, which is why we emphasize the 2D geometry in the abstract. In fact, for a local interaction $V(r)$, the condition for a bound state is $\int V(r)rdr<0$ in 2D whereas this is not enough in 3D, see for example Simon, *Annals of Physics* (1976) or *Quantum mechanics* by Landau & Lifshitz. The case analyzed in the present paper is however more complicated since the induced interaction appearing in the Schrödinger equation is non-local as discussed below Eq. (13). We nevertheless expect the 2D geometry to stabilize bound states as compared to 3D. This is now explained more carefully in the beginning of Sec. 6 where we cite the book by Landau & Lifshitz.

>*18- Fig. 2, I advise to a add a vertical black line at x=0 position (no additional caption is needed).*

We agree with the referee and have changed Fig. 2 accordingly.

>*19- Fig. 2, the horizontal dashed line seems to correspond to 1/2 and it is not clear if this is a coincidence as typically 1/2 is not a special value for the residue.*

The referee is correct - the horizontal dashed line in the lower panel of Fig. 2 is close to $1/2$ with a small deviation determined by the interaction between bosons in BEC. This interesting feature of the upper polaron follows from Eq. (8a) since $A=1$ for an ideal BEC giving $Z=1/2$ when $n_0=1$. For the value $U_B/t_B=0.07$ used to make Fig. 2 one has $A=1.00284$ giving $Z=0.49929$. We now comment on this in the manuscript.

>*20- Fig. 4, Figures are too small, 0 and 1 values are not shown. The projection angle in (b) panel is not optimal. Alternatively, top view as in Fig. 5 can be tried. Try to improve the figures.*

We thank the referee for pointing out this problem and have made Fig. 4 larger. We do however respectfully prefer to keep the ranges of the vertical axis as is. The reason is that the residue in general does not reach 0 and 1 so that it would appear compressed making the plots harder to read if we chose to show a range between 0 and 1.

---

## Editorial Decision

published